# Rethinking Losses for Diffusion Bridge Samplers

**Sebastian Sanokowski**[1,2]     **Lukas Gruber** [2]     **Christoph Bartmann** [2]

**Sepp Hochreiter**[2,3]     **Sebastian Lehner**[2]

[1] Technical University Munich, Chair of Robotics, Artificial Intelligence and Embedded Systems
[2] ELLIS Unit Linz, LIT AI Lab, Johannes Kepler University Linz, Austria
[3] NXAI Lab & NXAI GmbH, Linz, Austria

`sebastian.sanokowski[at]tum.de`

## Abstract

Diffusion bridges are a promising class of deep-learning methods for sampling from unnormalized distributions. Recent works show that the Log Variance (LV) loss consistently outperforms the reverse Kullback-Leibler (rKL) loss when using the reparametrization trick to compute rKL-gradients. While the on-policy LV loss yields identical gradients to the rKL loss when combined with the log-derivative trick for diffusion samplers with non-learnable forward processes, this equivalence does not hold for diffusion bridges or when diffusion coefficients are learned. Based on this insight we argue that for diffusion bridges the LV loss does not represent an optimization objective that can be motivated like the rKL loss via the data processing inequality. Our analysis shows that employing the rKL loss with the log-derivative trick (rKL-LD) does not only avoid these conceptual problems but also consistently outperforms the LV loss. Experimental results with different types of diffusion bridges on challenging benchmarks show that samplers trained with the rKL-LD loss achieve better performance. From a practical perspective we find that rKL-LD requires significantly less hyperparameter optimization and yields more stable training behavior.[1]

## 1 Introduction

We consider the task of learning to generate samples $X \in \mathbb{R}^N$ from an unnormalized target distribution:

$$\pi_0(X) = \frac{\exp\left(-\mathcal{E}(X)\right)}{\mathcal{Z}} \quad \text{with} \quad \mathcal{Z} = \int_{\mathbb{R}^\mathbb{N}} \exp\left(-\mathcal{E}(X)\right) dX, \tag{1}$$

where $\mathcal{Z}$ denotes the partition function and $\mathcal{E} : \mathbb{R}^N \to \mathbb{R}$ is the energy function. In this setting, it is assumed that the energy function $\mathcal{E}$ of the target distribution can be evaluated and $\mathcal{Z}$ is unknown and computationally intractable. Importantly, there are no available samples from $\pi_0$. Sampling problems of this kind represent fundamental challenges in computational physics and chemistry and in Bayesian inference [Noé and Wu, 2018, Wu et al., 2019, Shih and Ermon, 2020]. Recent approaches have focused on training probability distributions parameterized by neural networks to approximate target distributions. Early deep learning-based methods explored exact likelihood models such as normalizing flows [Noé and Wu, 2018] and autoregressive models [Wu et al., 2019, Nicoli et al., 2020], while more recent work has turned to approximate likelihood models like diffusion models in continuous [Zhang and Chen, 2022, Berner et al., 2022] and discrete domains [Sanokowski et al.,

---

[1]Our code is available at `https://github.com/sanokows/RethinkingLossesforDiffusionBridgeSamplers`.

39th Conference on Neural Information Processing Systems (NeurIPS 2025).

2024, 2025]. In the continuous domain, initial research explored these so-called diffusion samplers that aim at reversing the forward time evolution processes that are given by the variance-preserving and variance-exploding stochastic differential equations (SDEs) [Zhang and Chen, 2022, Vargas et al., 2023, Berner et al., 2022]. More recently, diffusion samplers based on diffusion bridges employ learnable forward processes and have established a new state-of-the-art in the field [Richter and Berner, 2024, Vargas et al., 2024, Blessing et al., 2025]. These models are typically trained using either **(i)** the reverse Kullback-Leibler divergence loss with gradients from the reparameterization trick (rKL-R) or **(ii)** the Log Variance (LV) loss where backpropagation through an expectation is not necessary [Richter and Berner, 2024]. However, diffusion bridge samplers face significant practical challenges. The rKL-R loss is susceptible to vanishing and exploding gradient problems when employing numerous diffusion steps [Zhang and Chen, 2022, Vargas et al., 2023] and has been empirically shown to underperform compared to the LV loss. Conversely, diffusion bridge samplers trained with the LV loss are prone to training instabilities and thus need extensive hyperparameter tuning.

In this work, we make three key contributions to address these limitations of diffusion bridge samplers. **(i)** We identify crucial problems in the application of the popular Log Variance (LV) loss in diffusion bridges. While the gradients of the LV loss and the gradients of the reverse KL divergence, when optimized with the log-derivative trick (rKL-LD), are identical when only the reverse diffusion control is learned, this equivalence does not hold in the context of diffusion bridge samplers or when diffusion coefficients are learned. We demonstrate, in this case, the LV loss suffers from training instabilities and is based on a divergence that does not satisfy the data processing inequality which represents an important theoretical criterion for losses of latent variable models. **(ii)** Instead of using the LV loss, we advocate for training diffusion bridge samplers with the rKL-LD loss and show that it consistently outperforms diffusion samplers trained with LV and rKL-R losses. Additionally, we find that it is significantly less sensitive to hyperparameter tuning. **(iii)** In addition to the usual learned drift terms of SDEs we motivate learnable diffusion terms that enable dynamic adaptation of the exploration-exploitation trade-off and show that combined with the rKL-LD loss they yield significantly improved performance when applied to hitherto state-of-the-art samplers that build on diffusion bridges.

## 2 Problem Description

### 2.1 Diffusion Bridges

Diffusion models are generative models that learn to transport samples $X_T \in \mathbb{R}^N$ from a simple prior distribution $\pi_T$ to samples that are distributed according to a target distribution $X_0 \sim \pi_0$. This transport map is defined by the reverse diffusion process, whose parameters are learned with the objective of inverting a forward diffusion process. The parameters of the corresponding forward SDE are either learned or predefined. Examples for later case are the variance-exploding or variance-preserving SDEs Song et al. [2021]. For these predefined forward SDEs the selection of parameters—particularly diffusion coefficients must be carefully adjusted since they determine the stochastic process which the diffusion model needs to revert in order to transport from $\pi_T$ to $\pi_0$. In diffusion bridges, however, also the forward SDE is learnable. These models are thus equipped with the flexibility to freely learn which transport path to use between $\pi_T$ to $\pi_0$. Due to this flexibility diffusion bridges have recently attracted increased research interest and represent the state-of-the-art in a wide range of applications, including the data generation and pairing problem [De Bortoli et al., 2021, 2024] or the problem of sampling from unnormalized distributions, which is the focus of this paper.

In diffusion bridges, the forward diffusion process is defined by the following SDE:

$$dX_t = f_\phi(X_t, t)\, dt + \sigma_t\, dW_t \quad \text{where} \quad t \in [0, T], \quad X_0 \sim \pi_0, \tag{2}$$

where $W_t \in \mathbb{R}^N$ is a Brownian motion, i.e. $dW_t = W_{t+dt} - W_t \sim \mathcal{N}(0, I_N\, dt)$ which is multiplied with diffusion coefficients $\sigma_t \in \mathbb{R}^N$, and $\phi$ are the learnable parameters of the forward diffusion control $f_\phi(X_t, t) \in \mathbb{R}^N$. The reverse process is defined as:

$$dX_\tau = r_\alpha(X_\tau, \tau)\, d\tau + \sigma_\tau\, dW_\tau \quad \text{where} \quad \tau \in [0, T], \quad X_{\tau=0} \sim \pi_T, \tag{3}$$

with learnable reverse control $r_\alpha(X_\tau, \tau) \in \mathbb{R}^N$. Here $\alpha$ denotes learnable parameters of the reverse diffusion process, which evolves in the opposite time direction $t = T - \tau$.

In practice, the reverse process is often simulated via Euler-Maruyama integration with step size $\Delta\tau = \Delta t \geq 0$ for $T$ steps so that the reverse SDE generation process is given by:

$$g_\alpha(X_\tau, \epsilon_\tau, \tau) = X_{\tau+1} = X_\tau + r_\alpha(X_\tau, \tau)\Delta\tau + \sigma_\tau\sqrt{\Delta\tau}\,\epsilon_\tau. \tag{4}$$

The conditional probability for a step in the reverse direction is then given by:

$$q_\alpha(X_{t-1}|X_t) = \mathcal{N}(X_{t-1}; X_t + r_\alpha(X_t, t)\,\Delta t, \sigma_t^2\Delta t) \tag{5}$$

and for a step in the forward direction by:

$$p_\phi(X_t|X_{t-1}) = \mathcal{N}(X_t; X_{t-1} + f_\phi(X_{t-1}, t-1)\,\Delta t, \sigma_{t-1}^2\Delta t). \tag{6}$$

## 2.2 Parameterization of Diffusion Bridge Samplers

In the following, we will introduce two common parameterizations of diffusion bridge samplers with a focus on the distinction between SDE parameters that are shared or non-shared between the forward and reverse processes. These considerations will reveal the conceptual problem of the LV loss and motivate the introduction of diffusion sampler training via rKL-LD. For this purpose we denote with $\phi$ and $\alpha$ the parameters of the forward and reverse process, respectively, that are non-shared between the two processes. The parameters that are shared are denoted via $\nu$. We denote the entirety of the parameters of a diffusion bridge as $\theta = (\alpha, \phi, \nu)$. This formulation encompasses two popular diffusion bridge samplers: Denoising Bridge Samplers (DBS) and Controlled Monte Carlo Diffusion (CMCD).

**Denoising Bridge Samplers:** DBS, introduced by Richter and Berner [2024], uses separate neural networks for reverse drift $r_\alpha$ and forward drift $f_\phi$ with non-shared parameters. In the underdamped DBS variant proposed in Blessing et al. [2025] additional SDE-related parameters including diffusion coefficients are learned, resulting in $\nu \neq \varnothing$.

**Controlled Monte Carlo Diffusion:** CMCD, introduced by Vargas et al. [2024], parameterizes both reverse and forward diffusion processes with the same parameterized control $u_\gamma$ with shared parameters $\gamma$. In CMCD, all learnable parameters are shared, i.e. $\phi = \alpha = \varnothing$. The control terms of the forward and reverse diffusion processes are guided by the score $\nabla_X \log \pi_t(X)$ of an interpolation between the forward and reverse process: $\pi_t(X_t) = \pi_0(X_t)^{\eta(t)}\pi_T(X_t)^{1-\eta(t)}$, where $\eta(t) \in [0,1]$ is a learned monotonically decreasing function with $\eta(0) = 1$ and $\eta(T) = 0$ (see App. D.3 for more details). The control of the reverse diffusion processes in Eq. 3 is then defined by:

$$r_\gamma(X_\tau, \tau) = \frac{\sigma_\tau^2}{2}\nabla_{X_\tau}\log\pi_\tau(X_\tau) + u_\gamma(X_\tau, \tau)$$

and the control of the forward diffusion process in Eq. 2 by:

$$f_\gamma(X_t, t) = \frac{\sigma_t^2}{2}\nabla_{X_t}\log\pi_t(X_t) - u_\gamma(X_t, t).$$

CMCD is initialized such that $u_\gamma(X_t, t) = 0$, which ensures that before training, the simulation of the reverse diffusion process Eq. 4 is equivalent to the Unadjusted Langevin Annealing. Intuitively, CMCD learns to modify unadjusted Langevin dynamics with an additive control term so that the resulting process efficiently transports samples from the prior to the target distribution within a limited number of diffusion steps $T$. In CMCD, the parameter structure can be represented by $\nu = (\gamma, \{\eta_t, \sigma_t\}_{t=1,...,T})$ when also diffusion coefficients are learned.

## 2.3 Training of Diffusion Samplers

The loss to train diffusion models is typically based on a divergence between the underlying marginal distribution $q_{\alpha,\nu}(X_0)$ induced by the reverse diffusion process and the target distribution $\pi_0(X_0)$. In this context, $f$-divergences are a popular choice [Csiszár, 1967]:

$$D_f(P(X)\,\|\,Q(X)) = \int Q(X)f\left(\frac{P(X)}{Q(X)}\right)dX,$$

where $f$ is a convex function satisfying $f(1) = 0$ and $P$ and $Q$ are two probability distributions satisfying $P << Q$, i.e. $P$ is absolutely continuous with respect to $Q$. The choice of $f$ significantly influences the learning dynamics in terms of mode-seeking vs. mass-covering properties.

**Reverse KL Divergence:** The rKL corresponds to $f(t) = -\log(t)$ and is given by:

$$D_{KL}(q_{\alpha,\nu}(X) \| \pi_0(X)) = \mathbb{E}_{X \sim q_{\alpha,\nu}(X)} \left[ \log \frac{q_{\alpha,\nu}(X)}{\pi_0(X)} \right]. \tag{7}$$

The rKL divergence is a convenient choice for learning to sample from unnormalized target distributions as the expectations are calculated over model samples $X \sim q_{\alpha,\nu}(X)$. However, the rKL exhibits a mode-seeking behavior, i.e. $q_{\alpha,\nu}$ tends to focus on the high-density regions of the target distribution $\pi_0$, potentially ignoring low-density regions or modes with smaller probability mass which leads to a bias between the target and variational distribution that cannot be corrected with Neural Importance Sampling or Neural Markov-Chain-Monte-Carlo [Nicoli et al., 2020].

**Forward KL Divergence:** The fKL corresponds to $f(t) = t\log(t)$ and is given by:

$$D_{KL}(\pi_0(X) \| q_{\alpha,\nu}(X)) = \mathbb{E}_{X \sim \pi_0(X)} \left[ \log \frac{\pi_0(X)}{q_{\alpha,\nu}(X)} \right]. \tag{8}$$

In contrast to the rKL it is mass-covering, i.e. $q_{\alpha,\nu}$ tends to cover the entire support of $\pi_0$, even at the cost of placing high probability in low-density regions. While this mass covering property is useful for sampling applications, because it enables asymptotically unbiased estimates of observables it comes at the cost of sample efficiency, and the fKL divergence cannot be straightforwardly used as a loss function as samples from $X \sim \pi_0(X)$ are not available. This problem can, however, be mitigated by either rewriting the expectation with respect to samples from $\pi_0(X)$ to an expectation over samples from $q_{\alpha,\nu}(X)$ by incorporating self-normalized importance weights as done in Müller et al. [2019], Jing et al. [2022], Midgley et al. [2023], Sanokowski et al. [2025].

**Jeffrey Distance:** Another possible choice is the Jeffrey distance which corresponds to $f(t) = (t-1)\log(t)$. This yields:

$$D_{\text{Jeff}}(\pi_0(X), q_{\alpha,\nu}(X)) = D_{KL}(\pi_0(X) \| q_{\alpha,\nu}(X)) + D_{KL}(q_{\alpha,\nu}(X) \| \pi_0(X)). \tag{9}$$

This divergence is a sum of the rKL and fKL and is therefore a trade-off between the mode-seeking and mass-covering behavior of the rKL and fKL divergence. In the context of sampling this loss is for example used in Noé and Wu [2018].

**Data Processing Inequality:** For expressive latent variable models like diffusion samplers, the marginal variational distribution $q_{\alpha,\nu}(X_0)$ is typically intractable. Therefore, losses for diffusion models are in practice often motivated by the data processing inequality for $f$-divergences that satisfy the following monotonicity relation [Zhang and Chen, 2022, Vargas et al., 2023, Sanokowski et al., 2024, Blessing et al., 2025]:

$$D_f(\pi_0(X_0) \| q_{\alpha,\nu}(X_0)) \le D_f(p_{\phi,\nu}(X_{0:T}) \| q_{\alpha,\nu}(X_{0:T})). \tag{10}$$

The right-hand side of this inequality is (up to a normalization constant) tractable for diffusion bridges as it is based on joint distributions of the diffusion paths $X_{0:T}$. Using Eq. 5 and Eq. 6 the corresponding joint probability distributions can be efficiently calculated with:

$$p_{\phi,\nu}(X_{0:T}) = \pi_0(X_0) \prod_{t=1}^{T} p_{\phi,\nu}(X_t | X_{t-1}), \quad q_{\alpha,\nu}(X_{0:T}) = \pi_T(X_T) \prod_{t=1}^{T} q_{\alpha,\nu}(X_{t-1} | X_t).$$

The parameters $(\alpha, \phi, \nu)$ of the diffusion bridge (Sec. 2.2) are optimized with the objective of minimizing the right-hand side of Eq. 10. This training objective corresponds to maximizing the Evidence Lower Bound (ELBO) in latent variable models [Blessing et al., 2024]. Eq. 10 states explicitly that any reduction of the right-hand side leads to a tighter bound on the divergence between the intractable marginals on the left-hand side. In fact it is well known in generative modeling that diffusion losses that are based on properly bounded objectives yield better sample likelihoods [Song et al., 2021]. We therefore argue that the data processing inequality is an essential property for losses in sampling setting and that losses that do not provide such a conceptual footing might be practically problematic (see Sec. 2.3.1).

### 2.3.1 Log Variance Loss

Richter et al. [2020] propose the LV loss as a convenient replacement of the rKL-based loss functions in Bayesian variational inference and it has since then been frequently used to train diffusion

samplers [Richter and Berner, 2024, Vargas et al., 2024, Sendera et al., 2024, Chen et al., 2024a, Blessing et al., 2024, He et al., 2025]. In the context of GflowNets Bengio et al. [2021] this loss is also known as the Trajectory Balance loss. For diffusion bridges, i.e. when both the reverse process $q_{\alpha,\nu}$ and the forward process $p_{\phi,\nu}$ involve learnable parameters the LV takes the following form:

$$D_{LV}^{\omega}(q_{\alpha,\nu}(X_{0:T}) \,\|\, p_{\phi,\nu}(X_{0:T})) = \frac{1}{2}\mathbb{E}_{X_{0:T}\sim\omega}\left[\left(\log\frac{q_{\alpha,\nu}(X_{0:T})}{p_{\phi,\nu}(X_{0:T})} - b_{\alpha,\phi,\nu}^{\omega}\right)^2\right], \qquad (11)$$

where $\omega$ is a suitable proposal distribution and $b_{\alpha,\phi,\nu}^{\omega} = \mathbb{E}_{X_{0:T}\sim\omega}\left[\log\frac{q_{\alpha,\nu}(X_{0:T})}{p_{\phi,\nu}(X_{0:T})}\right]$. If $\omega$ contains the support of $q_{\alpha,\nu}$ and $p_{\phi,\nu}(X_{0:T})$ the LV loss is zero if and only if $q_{\alpha,\nu}(X_{0:T}) = p_{\phi,\nu}(X_{0:T})$. In the simplest case $\omega = \text{stop\_gradient}(q_{\alpha,\nu}) \coloneqq q_{\alpha,\nu}^*$, for which we will refer to as the on-policy Log Variance (OP-LV) loss. For diffusion samplers, the general LV loss and the OP-LV loss were recently proposed in Richter and Berner [2024], however, as also noted in Malkin et al. [2022] the LV loss is not an $f$-divergence and we show via the following simple counter-example that the LV loss does not satisfy the data processing inequality.

---

**Violation of the Data Processing Inequality by the Log Variance Loss**

Consider the following distributions on the unit square $\Omega = [0,1]^2$:

- $p(x,y) = 1$ (uniform distribution over $\Omega$),

- $q_n(x,y) = \begin{cases} \frac{2n}{n+1} & \text{if} \quad x+y < 1 \\ \frac{2}{n+1} & \text{if} \quad x+y \geq 1, \end{cases}$

where $n \in \mathbb{R}_{>0}$ determines the ratio between the amount of probability mass located below and above the diagonal. For this choice, both $\text{Var}_{x\sim q_n}\left[\log\frac{q_n(x)}{p(x)}\right]$ and $\text{Var}_{(x,y)\sim q_n}\left[\log\frac{q_n(x,y)}{p(x,y)}\right]$ can be computed analytically (see App. A.5).
As illustrated below, these calculations show that for $n < 10^{-2}$ the Log Variance loss does not satisfy the data processing inequality (DPI):

$$\text{Var}_{x\sim q_n}\left[\log\frac{q_n(x)}{p(x)}\right] \not\leq \text{Var}_{(x,y)\sim q_n}\left[\log\frac{q_n(x,y)}{p(x,y)}\right],$$

where the joint distribution $q_n(x,y)$ is absolutely continuous with respect to $p(x,y)$, and vice versa.

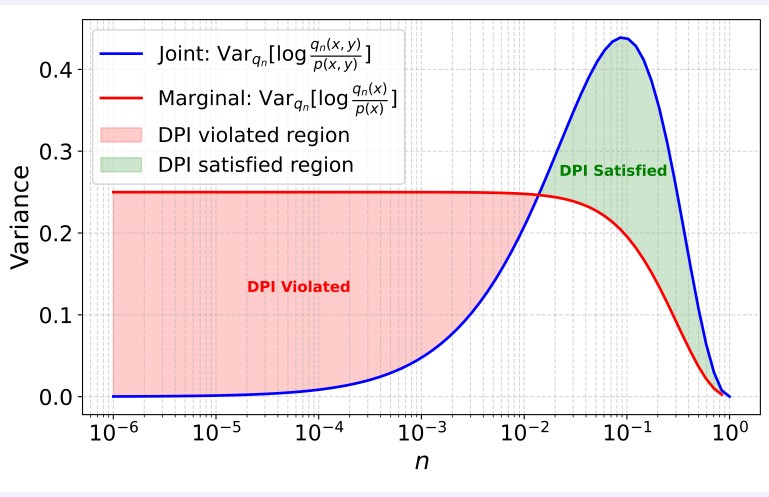

---

Consequently, we argue that its application to latent variable models is potentially problematic since it conflicts with the rationale of diffusion sampler training based on divergences of joint probabilities motivated by the data processing inequality (see Sec. 2.3). We provide an additional counter example in discrete state spaces is proved in App. A.5.

## 3 Method

### 3.1 Reverse KL Loss with Log-derivative Trick and Control Variate

One way to compute the gradient of the rKL loss is to use the reparametrization trick [Kingma and Welling, 2014] as explained in more detail in App. A.1. In diffusion samplers, the repeated application of the reparametrization trick is, however, likely to give rise to the vanishing or exploding gradient problem, which might explain the suboptimal behavior of rKL loss when minimized with the usage of the reparametrization trick [Zhang and Chen, 2022, Vargas et al., 2023]. Several recent works demonstrate that the LV loss outperforms the rKL loss with the reparametrization trick [Richter and Berner, 2024, Vargas et al., 2024, Chen et al., 2024a]. It is argued that this is due to the mode-seeking tendency associated with the rKL objective which results in mode collapse. While the aforementioned works applied the reparametrization trick in conjunction with the rKL objective we investigate the rKL objective with the log-derivative gradient estimator. The gradients corresponding to the rKL-LD with respect to the parameters $(\alpha, \phi, \nu)$ read (App. A.4):

$$\nabla_\alpha^{LD, q_{\alpha,\nu}} D_{KL}(q_{\alpha,\nu} \| p_{\phi,\nu}) = \mathbb{E}_{X_{0:T} \sim q_{\alpha,\nu}} \left[ \left( \log \frac{q_{\alpha,\nu}(X_{0:T})}{p_{\phi,\nu}(X_{0:T})} - b_{\alpha,\phi,\nu}^{q_{\alpha,\nu}} \right) \cdot \nabla_\alpha \log q_{\alpha,\nu}(X_{0:T}) \right]$$

$$\nabla_\phi D_{KL}(q_{\alpha,\nu} \| p_{\phi,\nu}) = -\mathbb{E}_{X_{0:T} \sim q_{\alpha,\nu}} \left[ \nabla_\phi \log p_{\phi,\nu}(X_{0:T}) \right] \qquad (12)$$

$$\nabla_\nu^{LD, q_{\alpha,\nu}} D_{KL}(q_{\alpha,\nu} \| p_{\phi,\nu}) = \mathbb{E}_{X_{0:T} \sim q_{\alpha,\nu}} \left[ \left( \log \frac{q_{\alpha,\nu}(X_{0:T})}{p_{\phi,\nu}(X_{0:T})} - b_{\alpha,\phi,\nu}^{q_{\alpha,\nu}} \right) \nabla_\nu \log q_{\alpha,\nu}(X_{0:T}) \right]$$
$$- \mathbb{E}_{X_{0:T} \sim q_{\alpha,\nu}} \left[ \nabla_\nu \log p_{\phi,\nu}(X_{0:T}), \right]$$

where $\nabla_\theta^{LD, \omega_\theta}$ is an operator that computes the gradient of parameters $\theta$ with respect the expectation over a distribution $\omega_\theta$ by applying the log-derivative trick and by reducing the variance with a control variate $b_\theta^{\omega_\theta}$ as defined in App. A.2.

### 3.2 Gradient of the Log Variance Loss

The corresponding gradients of the LV loss with respect to bridge parameters $(\alpha, \phi, \nu)$ are given by (see App. A.3):

$$\begin{pmatrix} \nabla_\alpha \\ \nabla_\phi \\ \nabla_\nu \end{pmatrix} D_{LV}^\omega(q_{\alpha,\nu}, p_{\phi,\nu}) = \begin{pmatrix} \mathbb{E}_{X_{0:T} \sim \omega} \left[ \left( \log \frac{q_{\alpha,\nu}(X_{0:T})}{p_{\phi,\nu}(X_{0:T})} - b_{\alpha,\phi,\nu}^\omega \right) \cdot \nabla_\alpha \log q_{\alpha,\nu}(X_{0:T}) \right] \\ -\mathbb{E}_{X_{0:T} \sim \omega} \left[ \left( \log \frac{q_{\alpha,\nu}(X_{0:T})}{p_{\phi,\nu}(X_{0:T})} - b_{\alpha,\phi,\nu}^\omega \right) \cdot \nabla_\phi \log p_{\phi,\nu}(X_{0:T}) \right] \\ \mathbb{E}_{X_{0:T} \sim \omega} \left[ \left( \log \frac{q_{\alpha,\nu}(X_{0:T})}{p_{\phi,\nu}(X_{0:T})} - b_{\alpha,\phi,\nu}^\omega \right) \cdot \nabla_\nu \log \frac{q_{\alpha,\nu}(X_{0:T})}{p_{\phi,\nu}(X_{0:T})} \right] \end{pmatrix} \qquad (13)$$

The gradient of the LV loss is related to well-known $f$-divergences in the following way. With the choice $\omega = \text{stop\_gradient}(q_{\alpha,\nu}) =: q_{\alpha,\nu}^*$ one obtains:

$$\nabla_\alpha D_{LV}^{q_{\alpha,\nu}^*}(q_{\alpha,\nu}, p_{\phi,\nu}) = \nabla_\alpha^{LD, q_{\alpha,\nu}} D_{KL}(q_{\alpha,\nu} \| p_{\phi,\nu}),$$

Hence, for non-shared parameters of the reverse diffusion process $\alpha$ the gradient estimator of the LV loss coincides with the gradient of the rKL divergence computed with the log-derivative trick.

When the off-policy distribution is $\omega = \text{stop\_gradient}(p_{\phi,\nu}) =: p_{\phi,\nu}^*$ the gradient with respect of $\phi$ is given by:

$$\nabla_\phi D_{LV}^{p_{\phi,\nu}^*}(q_{\alpha,\nu}, p_{\phi,\nu}) = \nabla_\phi^{LD, p_{\phi,\nu}} D_{KL}(p_{\phi,\nu} \| q_{\alpha,\nu})$$

and therefore exactly the same as the gradient of the forward KL divergence with log-derivative trick. For the OP-LV loss, however, we have at initialization $|p_{\phi,\nu}^* - q_{\alpha,\nu}^*| \gg 0$ and therefore $\nabla_\phi D_{LV}^{p_{\phi,\nu}^*}(q_{\alpha,\nu}, p_{\phi,\nu})$ can be seen as a biased estimate of $\nabla_\phi^{LD, p_{\phi,\nu}} D_{KL}(p_{\phi,\nu} \| q_{\alpha,\nu})$ as it can alternatively be estimated with the usage of Neural Importance Sampling in the following way:

$$\nabla_\phi^{LD, p_{\phi,\nu}} D_{KL}(p_{\phi,\nu} \| q_{\alpha,\nu}) = -\mathbb{E}_{X_{0:T} \sim q_{\alpha,\nu}} \left[ w(X_{0:T}) \left( \log \frac{q_{\alpha,\nu}(X_{0:T})}{p_{\phi,\nu}(X_{0:T})} - b_{\alpha,\phi,\nu}^{q_{\alpha,\nu}} \right) \cdot \nabla_\phi \log p_{\phi,\nu}(X_{0:T}) \right]$$

where (with abuse of notation) $w(X_{0:T}) = \frac{N}{\Omega} \frac{p_{\phi,\nu}(X_{0:T})}{q_{\alpha,\nu}(X_{0:T})}$ with $\Omega = \sum_{i=1}^N \frac{p_{\phi,\nu}(X_{0:T,i})}{q_{\alpha,\nu}(X_{0:T,i})}$ are self normalized importance weights. However, as $q_{\alpha,\nu}$ approximates $p_{\phi,\nu}$ and therefore $w(X_{0:T}) \to 1$, this bias is continually reduced.

For parameters that are shared between the forward and backward diffusion process $\nu$ we have for OP-LV at optimality when $p^*_{\phi,\nu} = q^*_{\alpha,\nu}$

$$\nabla_\nu D^{p^*_{\phi,\nu}}_{\text{LV}}(q_{\alpha,\nu}, p_{\phi,\nu}) = \nabla_\nu^{LD,p_{\phi,\nu}} D_{KL}(p_{\phi,\nu} \| q^*_{\alpha,\nu}) + \nabla_\nu^{LD,q_{\alpha,\nu}} D_{KL}(q_{\alpha,\nu} \| p^*_{\phi,\nu}),$$

where the right-hand side is related to the gradient of the Jeffrey distance with a $\text{stop\_gradient}$ operation in the second argument of each KL divergence. Therefore, at initialization when $|p^*_{\phi,\nu} - q^*_{\alpha,\nu}| \gg 0$ the gradient is biased as $w(X_{0:T}) \neq 1$ and additionally there is a difference to the gradient of the Jeffrey distance due to the $\text{stop\_gradient}$ operations.

A comparison of these gradients with the gradients of the rKL-LD loss in Eq. 12 shows that when $\phi = \nu = \varnothing$ the gradients of the OP-LV loss and rKL-LD loss are identical. Conversely, when $\omega \neq \text{stop\_gradient}(q_\theta)$, these two losses yield, in general, different gradients. Similarly, when $\phi \neq \varnothing, \nu \neq \varnothing$, the gradients with respect to these parameters are different, in which case an evaluation of the rKL-LD loss has not been considered in recent literature. These considerations and the fact that the LV loss violates the data processing inequality put the validity of the LV loss for diffusion bridge samplers in question and in fact we observe in our experiments that the LV loss often exhibits unstable behavior and requires hyperparameters to be carefully tuned (see Sec. 5). Additionally, we provide in Sec. 5 a detailed comparison between the LV and rKL-LD loss for DBS and CMCD, where we observe that rKL-LD yields better diffusion bridge samplers than the LV loss.

### 3.3 Learning of SDE parameters

SDEs in diffusion samplers are governed by diffusion coefficients that significantly influence their sampling performance. While previous work has predominantly focused on non-learned diffusion coefficients, the choice of these parameters involves nuanced considerations that can strongly affect the model's ability to capture complex distributions. In this section, we motivate why it is beneficial to learn these coefficients.

The diffusion coefficients control the amount of stochasticity injected throughout the sampling process and thereby regulate how broadly the model explores the data space. Larger coefficients increase randomness, allowing the sampler to better cover the support of high-entropy or multimodal target distributions. However, excessive noise reduces the signal-to-noise ratio and can obscure important structural information in the data, leading to poorer reconstructions and slower convergence. Conversely, too little noise restricts exploration and may cause the sampler to miss significant regions of the target distribution.

This interplay between coverage and fidelity reveals a fundamental trade-off in selecting appropriate diffusion coefficients. Learning these coefficients rather than fixing them a priori allows the model to adaptively tune the level of stochasticity across time and dimensions, achieving a better balance between exploration and precision. This trade-off motivates the introduction of learnable diffusion coefficients. The optimal magnitude of noise injection varies across different problems and even across different dimensions of the same problem, suggesting that adaptively learning these parameters can lead to more efficient approximations [Blessing et al., 2025]. In our experiments in Sec. 5, we demonstrate that diffusion coefficient learning yields significantly better performance in combination with the rKL-LD loss. While we observe that diffusion coefficient learning consistently improves results with rKL-LD it tends to worsen performance and increases hyperparameter sensitivity when using the LV loss, highlighting the benefit of rKL-LD.

## 4 Related Work

For continuous diffusion samplers, the rKL loss is typically used with the reparametrization trick [Vargas et al., 2023, Berner et al., 2022]. More recently, Richter and Berner [2024] proposed the LV loss as an alternative and showed that it outperforms the rKL loss with the reparametrization trick. The corresponding experiments are performed with the Path Integral Sampler [Zhang and Chen, 2022] and the Time-Reversed Diffusion Sampler [Berner et al., 2022], in which the forward diffusion process is not learned. However, they report that the application of LV to diffusion bridges results in poor performance and numerical instabilities. In Vargas et al. [2024], Chen et al. [2024a] the LV loss is employed in conjunction with diffusion bridges and both works report that it outperforms the rKL loss with the parametrization trick. Frequently, the mode-collapse tendency of rKL is given as

an explanation for its inferior performance in the context of diffusion samplers [Richter and Berner, 2024]. The rKL with log-derivative trick has recently been successfully applied to diffusion samplers in discrete domains. Examples of such problems arise in combinatorial optimization and statistical physics of spin lattices [Sanokowski et al., 2024, 2025]. Some of the considerations in Sec. 3.2 are also made in the context of GflowNets in Malkin et al. [2022].

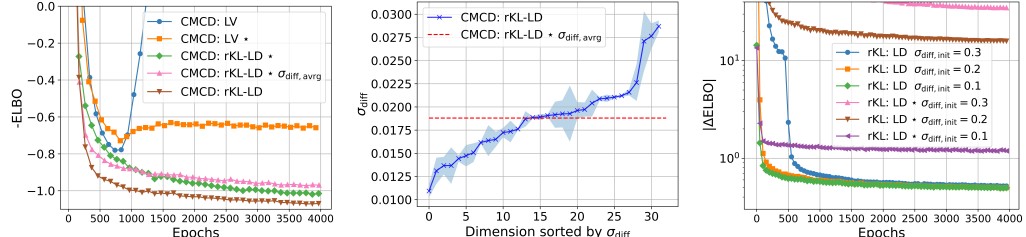

Figure 1: Models with fixed $\sigma_{\mathrm{diff}}$ are marked with $\star$. Left: Training curves on the Brownian task of CMCD trained with LV loss and rKL-LD loss. Middle: Plot of the learned $\sigma_{\mathrm{diff}}$ in ascending order of the CMCD-rKL-LD run from the left figure. Right: Training curves on the Seeds task, where CMCD-rKL-LD $\sigma_{\mathrm{diff,init}}$ is compared to CMCD-rKL-LD $\star$ $\sigma_{\mathrm{diff,init}}$ at different initializations of $\sigma_{\mathrm{diff}}$.

| ELBO ($\uparrow$) | Seeds (26d) | Sonar (61d) | Credit (25d) | Brownian (32d) | LGCP (1600d) |
|---|---|---|---|---|---|
| **CMCD: rKL-R** $\star$ | $-74.37_{\pm 0.01}$ | $-109.69_{\pm 0.063}$ | $-504.99_{\pm 0.016}$ | $-0.30_{\pm 0.018}$ | $\mathbf{471.91_{\pm 0.291}}$ |
| **CMCD: LV** $\star$ | $-74.13_{\pm 0.01}$ | $-109.53_{\pm 0.062}$ | $-504.91_{\pm 0.002}$ | $-0.05_{\pm 0.002}$ | $460.84_{\pm 0.099}$ |
| **CMCD: LV** | $-73.53_{\pm 0.01}$ | $-109.66_{\pm 0.015}$ | $-628.39_{\pm 32.907}$ † | $-6.05_{\pm 0.028}$ † | $447.74_{\pm 0.0}$ † |
| **CMCD: rKL-LD** $\star$ | $-74.10_{\pm 0.01}$ | $-109.25_{\pm 0.007}$ | $-504.88_{\pm 0.003}$ | $0.36_{\pm 0.001}$ | $466.73_{\pm 0.027}$ |
| **CMCD: rKL-LD** | $\mathbf{-73.45_{\pm 0.01}}$ | $\mathbf{-108.83_{\pm 0.005}}$ | $\mathbf{-504.58_{\pm 0.001}}$ | $\mathbf{1.06_{\pm 0.0}}$ | $465.80_{\pm 0.02}$ |
| **DBS: rKL-R** $\star$ | $-75.09_{\pm 0.113}$ | $-119.58_{\pm 1.388}$ | $-528.15_{\pm 0.06}$ | $-3.13_{\pm 0.037}$ | $461.09_{\pm 0.112}$ |
| **DBS: LV** $\star$ | $-74.12_{\pm 0.005}$ | $-110.66_{\pm 0.013}$ | $-506.21_{\pm 0.109}$ | $-9.39_{\pm 0.028}$ † | $460.48_{\pm 1.408}$ |
| **DBS: LV** | $-74.14_{\pm 0.014}$ † | $-111.14_{\pm 0.022}$ | $-573.23_{\pm 0.374}$ † | $-9.39_{\pm 0.035}$ † | $455.49_{\pm 4.308}$ † |
| **DBS: rKL-LD** $\star$ | $-74.03_{\pm 0.002}$ | $-109.41_{\pm 0.013}$ | $-534.17_{\pm 0.427}$ | $-0.16_{\pm 0.016}$ | $469.73_{\pm 0.038}$ |
| **DBS: rKL-LD** | $\mathbf{-73.50_{\pm 0.0}}$ | $\mathbf{-108.88_{\pm 0.005}}$ | $\mathbf{-504.71_{\pm 0.017}}$ | $\mathbf{0.85_{\pm 0.006}}$ | $\mathbf{469.89_{\pm 0.039}}$ |

Table 1: Results on Bayesian learning benchmarks. The ELBO (the higher the better) is reported for various methods and tasks. $\star$ denotes that $\sigma_{\mathrm{diff}}$ is not learned during training. Divergent runs are denoted with †.

# 5 Experiments

| Task | GMM-40 (50d) | | | | MoS-10 (50d) | | | |
|---|---|---|---|---|---|---|---|---|
| Metric | Sinkhorn ($\downarrow$) | ELBO ($\uparrow$) | EMC ($\uparrow$) | MMD ($\downarrow$) | Sinkhorn ($\downarrow$) | ELBO ($\uparrow$) | EMC ($\uparrow$) | MMD ($\downarrow$) |
| **Ground Truth** | $875.21_{\pm 86.023}$ | $0.$ | $1.$ | $0.07_{\pm 0.001}$ | $449.06_{\pm 104.87}$ | $0.$ | $1.$ | $0.07_{\pm 0.001}$ |
| **CMCD: rKL-R** $\star$ | $50830.16_{\pm 3103.552}$ | $-37.37_{\pm 0.10}$ | $0.490_{\pm 0.201}$ | $1.77_{\pm 0.001}$ | $1605.24_{\pm 23.129}$ | $\mathbf{-19.88_{\pm 0.16}}$ | $0.628_{\pm 0.048}$ | $0.58_{\pm 0.006}$ |
| **CMCD: LV** $\star$ | $2559.20_{\pm 121.211}$ | $-37.37_{\pm 0.10}$ | $\mathbf{0.996_{\pm 0.001}}$ | $0.12_{\pm 0.0}$ | $1263.78_{\pm 50.713}$ | $-52.52_{\pm 0.70}$ | $0.971_{\pm 0.001}$ | $0.35_{\pm 0.002}$ |
| **CMCD: LV** | $2627.96_{\pm 130.8}$ | $-45.85_{\pm 0.17}$ | $\mathbf{0.996_{\pm 0.001}}$ | $0.13_{\pm 0.0}$ | $1181.82_{\pm 53.802}$ | $-43.63_{\pm 0.51}$ | $\mathbf{0.994_{\pm 0.001}}$ | $0.36_{\pm 0.002}$ |
| **CMCD: rKL-LD** $\star$ | $2362.47_{\pm 106.694}$ | $-26.78_{\pm 0.05}$ | $\mathbf{0.997_{\pm 0.001}}$ | $0.09_{\pm 0.001}$ | $\mathbf{915.91_{\pm 74.8}}$ | $-34.93_{\pm 0.25}$ | $0.981_{\pm 0.004}$ | $\mathbf{0.29_{\pm 0.003}}$ |
| **CMCD: rKL-LD** | $\mathbf{2301.16_{\pm 93.118}}$ | $\mathbf{-21.94_{\pm 0.10}}$ | $\mathbf{0.997_{\pm 0.000}}$ | $\mathbf{0.08_{\pm 0.0}}$ | $\mathbf{915.52_{\pm 74.62}}$ | $-34.93_{\pm 0.25}$ | $0.981_{\pm 0.004}$ | $\mathbf{0.29_{\pm 0.003}}$ |
| **DBS: rKL-R** $\star$ | $2569.60_{\pm 159.869}$ | $-80.49_{\pm 0.703}$ | $0.997_{\pm 0.000}$ | $0.12_{\pm 0.001}$ | $1908.32_{\pm 97.02}$ | $-33.87_{\pm 0.321}$ | $0.424_{\pm 0.032}$ | $0.43_{\pm 0.002}$ |
| **DBS: LV** $\star$ | $2073.09_{\pm 64.554}$ | $-35.45_{\pm 0.029}$ | $\mathbf{0.998_{\pm 0.000}}$ | $0.12_{\pm 0.0}$ | $1220.27_{\pm 47.066}$ | $-57.49_{\pm 0.375}$ | $0.980_{\pm 0.005}$ | $0.36_{\pm 0.002}$ |
| **DBS: LV** | $2067.56_{\pm 63.623}$ | $-42.27_{\pm 0.054}$ | $\mathbf{0.998_{\pm 0.000}}$ | $0.12_{\pm 0.0}$ | $1284.95_{\pm 54.832}$ | $-58.22_{\pm 0.398}$ | $0.961_{\pm 0.005}$ | $0.36_{\pm 0.002}$ |
| **DBS: rKL-LD** $\star$ | $2128.99_{\pm 69.267}$ | $-33.97_{\pm 0.035}$ | $\mathbf{0.998_{\pm 0.000}}$ | $0.12_{\pm 0.0}$ | $1052.31_{\pm 50.521}$ | $-43.67_{\pm 0.45}$ | $\mathbf{0.993_{\pm 0.001}}$ | $0.33_{\pm 0.004}$ |
| **DBS: rKL-LD** | $2133.50_{\pm 70.093}$ | $\mathbf{-30.44_{\pm 0.017}}$ | $\mathbf{0.998_{\pm 0.000}}$ | $\mathbf{0.11_{\pm 0.0}}$ | $1051.34_{\pm 50.545}$ | $-43.66_{\pm 0.448}$ | $\mathbf{0.993_{\pm 0.001}}$ | $0.33_{\pm 0.004}$ |

Table 2: Results on GMM-40 (50d) and MoS-10 (50d). Arrow $\downarrow$ denotes that lower values of the metric are better, and $\uparrow$ denotes that the higher values are the better. $\star$ denotes that $\sigma_{\mathrm{diff}}$ is not learned during training. Runs that diverge after reaching the reported value marked with †.

**Benchmarks:** Following Chen et al. [2024a], we evaluate our model on two types of tasks: In Tab. 1 we evaluate on Bayesian learning problems, where we report the ELBO due to the absence of data samples (see App. C). In Tab. 2 and Tab. 3 we evaluate Synthetic targets, where we report the Sinkhorn distance, Entropic Mode Coverage (EMC) Blessing et al. [2024] and Maximum Mean Discrepancy (MMD), which are all based on samples from the diffusion sampler and samples from

the target distribution (see App. C). On multimodal tasks, a combination of high ELBO and EMC values and low Sinkhorn and MMD distances indicate good performance. Detailed descriptions of all problem types are provided in App. C.1. To obtain a comprehensive evaluation of the compared loss functions, we assess their performance using both diffusion bridge sampling methods CMCD and DBS. Due to different parameter counts between the two methods, we do not aim for a fair comparison between CMCD and DBS. Our experimental setup mirrors Chen et al. [2024a], i.e. training all methods for 40.000 training iterations with a batch size of 2.000 on all tasks besides LGCP with a batch size of 300. Models are trained using 128 diffusion steps. We use a commonly used diffusion sampler architecture for all diffusion-based methods as described in App. D.3. In our experiments, we compare different variations of CMCD and DBS trained with different loss functions, denoted as rKL-R (Eq. 15), rKL-LD (Eq. 12), and LV (Eq. 13). We provide a pseudoalgorithm for each loss in App. D.4. For each loss, we report the results of a variation, where the diffusion coefficients $\sigma_{\text{diff}}$ of the underlying SDE are not learned. Whenever diffusion coefficients are not learned we denote it with a $\star$. For rKL-LD and LV we also report the results of the method when $\sigma_{\text{diff}}$ is learned. For each setting, we perform a grid search over the learning rates, the initial $\sigma_{\text{diff}}$ and the initial variance of a prior distribution $\sigma_{\text{prior}}$ as described in App. D.

**Ablation on Learnable Diffusion Coefficients and Divergent Behavior of Log Variance Loss:**
We first investigate the impact of learnable $\sigma_{\text{diff}} \in \mathbb{R}^N$ when combined with the rKL-LD loss, as shown in Fig. 1 (left and right). For simplicity we chose $\sigma_{\text{diff}}$ to be constant across time and leave time dependent $\sigma_{\text{diff}}$ up for future work. In Fig. 1 (left) we evaluate CMCD under several configurations on the Brownian Bayesian learning task (see App. C.1). Our baseline

| Task | Funnel (10d) | | Many Well (5d) | |
|---|---|---|---|---|
| Metric | Sinkhorn ($\downarrow$) | ELBO ($\uparrow$) | Sinkhorn ($\downarrow$) | ELBO ($\uparrow$) |
| **Ground Truth** | $64.77_{\pm 0.71}$ | $0.$ | $0.12_{\pm 0.0}$ | $-0.54$ |
| **CMCD: rKL-R** $\star$ | $113.38_{\pm 0.77}$ | $-2.46_{\pm 0.35}$ | $2.05_{\pm 0.052}$ | $-4.14_{\pm 0.001}$ |
| **CMCD: LV** $\star$ | $\mathbf{95.90_{\pm 2.58}}$ | $-0.67_{\pm 0.00}$ | $0.13_{\pm 0.0}$ | $-1.02_{\pm 0.001}$ |
| **CMCD: LV** | $102.94_{\pm 3.04}$ † | $-0.46_{\pm 0.01}$ † | $0.13_{\pm 0.0}$ † | $-1.40_{\pm 0.0}$ † |
| **CMCD: rKL-LD** $\star$ | $94.04_{\pm 2.239}$ | $-0.54_{\pm 0.01}$ | $0.12_{\pm 0.0}$ | $-0.75_{\pm 0.003}$ |
| **CMCD: rKL-LD** | $94.16_{\pm 2.55}$ | $\mathbf{-0.23_{\pm 0.01}}$ | $0.12_{\pm 0.0}$ | $\mathbf{-0.74_{\pm 0.003}}$ |
| **DBS: rKL-R** $\star$ | $111.12_{\pm 2.908}$ | $-0.69_{\pm 0.004}$ | $0.42_{\pm 0.001}$ | $-7.21_{\pm 0.005}$ |
| **DBS: LV** $\star$ | $335.12_{\pm 33.757}$ | $-2.66_{\pm 0.053}$ | $0.15_{\pm 0.006}$ † | $-6.27_{\pm 1.246}$ † |
| **DBS: LV** | $151.83_{\pm 8.647}$ | $-2.32_{\pm 0.111}$ | $0.20_{\pm 0.0}$ | $-2.92_{\pm 0.012}$ |
| **DBS: rKL-LD** $\star$ | $105.97_{\pm 2.377}$ | $-0.50_{\pm 0.003}$ | $0.12_{\pm 0.0}$ | $\mathbf{-0.65_{\pm 0.032}}$ |
| **DBS: rKL-LD** | $108.88_{\pm 3.44}$ | $-0.35_{\pm 0.001}$ | $0.12_{\pm 0.0}$ | $-0.65_{\pm 0.032}$ |

Table 3: Results on Funnel (10d) and Many Well (5d). Arrow $\downarrow$ denotes that lower values of the metric are better, and $\uparrow$ denotes that the higher values are the better. $\star$ denotes that $\sigma_{\text{diff}}$ is not learned during training. Divergent runs are marked with a †.

comparison is with LV loss, where $\sigma_{\text{diff}}$ are not updated during training. The LV loss is highly sensitive to the initial choices of $\sigma_{\text{prior}}$ and $\sigma_{\text{diff}}$ and often exhibits unstable behavior. When training $\sigma_{\text{diff}}$ with the LV loss, we observe divergent behavior across all hyperparameter choices. For comparison, we study our proposed loss function in two variants, with trainable and frozen $\sigma_{\text{diff}}$. For the frozen version we performed hyperparameter tuning on the initial value of $\sigma_{\text{diff}}$. The experimental results demonstrate that the rKL-LD loss consistently outperforms the LV loss across all tested configurations and that incorporating learnable diffusion coefficients further enhances model performance when using the rKL-LD loss. In Fig. 1 (middle), we analyze the learned $\sigma_{\text{diff}}$ across dimensions, showing their values in ascending order along with standard deviations computed from three independent seeds. The results reveal that $\sigma_{\text{diff}}$ systematically adopts different scales across dimensions while maintaining consistency between seeds. To test the hypothesis, whether different values of $\sigma_{\text{diff}}$ in each dimension are indeed beneficial, we additionally train a diffusion sampler with the rKL-LD loss with frozen sigma parameters initialized at the average $\sigma_{\text{diff}}$ of the best previously trained run (CMCD: rKL-LD $\star$ $\sigma_{\text{diff,avrg}}$ in Fig. 1 (left)). The results show that this uniform choice of $\sigma_{\text{diff}}$ across dimensions yields inferior results.
Fig. 1 (right) shows how the initial value of $\sigma_{\text{diff}}$ affects performance by comparing CMCD: rKL-LD with and without learned diffusion coefficients on the Seeds dataset. We track the convergence using $\Delta$ELBO, defined as $|\text{ELBO}_{\text{opt}} - \text{ELBO}|$, where we estimate $\text{ELBO}_{\text{opt}} = -73\,\text{nats}$ to enable visualization on a logarithmic scale. The results demonstrate that when CMCD learns $\sigma_{\text{diff}}$, it achieves much lower ELBO values regardless of the initial parameter choice. Without learned diffusion parameters, the model struggles to achieve low ELBO values.

**Results on Bayesian and Synthetic Targets:** Overall, the results in Tab. 1, 2, and Tab. 3 indicate that the divergent behavior when using the LV loss is present on most investigated benchmarks.

Our results further show that on Bayesian tasks in Tab. 1, when using the rKL-LD loss CMCD with frozen $\sigma_{\mathrm{diff}}$ significantly outperforms its counterpart trained with LV loss on all tasks, and the version trained with rKL-R on $4$ out of $5$ tasks. Similarly, DBS with frozen $\sigma_{\mathrm{diff}}$ in combination with rKL-LD significantly outperforms DBS variants trained with other losses in $4$ out of $5$ cases. If we additionally train $\sigma_{\mathrm{diff}}$, we observe that CMCD improves on $4$ out of $5$ tasks and DBS on all tasks when trained with rKL-LD. In contrast, we observe that learning $\sigma_{\mathrm{diff}}$ in combination with the LV loss deteriorates the performance of CMCD and of DBS in $4$ out of $5$ cases. In fact, learning $\sigma_{\mathrm{diff}}$ with the LV loss often leads to unstable learning dynamics as the runs diverge in $3$ out of $5$ cases for CMCD and in $4$ out of $5$ cases for DBS. On synthetic tasks in Tab. 2 and 3, CMCD with rKL-LD achieves the best Sinkhorn distance on MoS-10 (50d) and GMM-40 (50d) by a significant margin, and insignificantly better Sinkhorn distance than CMCD: LV ⋆ on Funnel. In terms of MMD, using the rKL-LD loss for CMCD significantly outperforms training with the other losses in all cases. In terms of ELBO, training CMCD with rKL-LD achieves better results than with LV on all $4$ synthetic tasks. For DBS we observe that rKL-LD ⋆ outperforms all variants of LV and rKL-R in terms of Sinkhorn in $3$ out of $4$ cases, in terms of ELBO in all $4$ cases and in terms of MMD in $1$ out of $2$ cases. On synthetic targets training $\sigma_{\mathrm{diff}}$ with DBS improves the method only significantly on GMM-40 (50d) in terms of ELBO and MMD, and on Funnel in terms of ELBO. However, learning $\sigma_{\mathrm{diff}}$ never detoriates the performance of DBS when trained with rKL-LD which is not the case for the LV loss. All methods except samplers trained with rKL-R ⋆ achieve on MoS-10 (50d) and GMM-40 (50d) an EMC value close to $1.$, indicating that all modes are covered. Furthermore, we extend experiments on GMM-40 and MoS-10 to a higher-dimensional setting and results in Tables 4 and 5 support our previous findings.

## 6 Conclusion and Limitations

In this study, we advocate for the training of diffusion bridge-based samplers using gradients of the reverse Kullback-Leibler divergence, estimated with the log-derivative trick (rKL-LD). Our analysis reveals an important insight: while the Log Variance (LV) loss and reverse KL loss are equivalent when training solely the reverse diffusion process, this equivalence does not hold when dealing with diffusion bridge samplers or learning diffusion coefficients. Furthermore, we show that the LV loss does not, in general, satisfy the data processing inequality, raising questions about its suitability for diffusion bridge samplers. Empirical results highlight the superiority of the proposed rKL-LD loss over the LV loss. Notably, employing the rKL-LD loss allows for further improvement of diffusion bridges by learning the diffusion coefficients, which also diminishes sensitivity to hyperparameter choices. We find that in contrast to the rKL-LD loss the LV loss frequently yields unstable training in particular when diffusion coefficients are learned. While the rKL-LD loss outperforms the LV loss and rKL loss with parametrization trick on a wide range of challenging sampling benchmarks, it remains susceptible to mode collapse when hyperparameters, particularly learning rates, are not sufficiently well tuned. Future research aimed at reducing the mode-collapsing tendency of rKL-based losses presents an interesting direction for further investigation. The LV loss has recently been combined with an off-policy sample buffer that incorporates MCMC updates and resampling strategies, which improves its stability [Sendera et al., 2024, Chen et al., 2024a]. A comprehensive comparison between the LV loss and rKL-LD loss in such a setting remains an important direction for future work.

## 7 Acknowledgements

The ELLIS Unit Linz, the LIT AI Lab, the Institute for Machine Learning, are supported by the Federal State Upper Austria. We thank the projects FWF AIRI FG 9-N (10.55776/FG9), AI4GreenHeatingGrids (FFG- 899943), Stars4Waters (HORIZON-CL6-2021-CLIMATE-01-01), FWF Bilateral Artificial Intelligence (10.55776/COE12). We thank NXAI GmbH, Audi AG, Silicon Austria Labs (SAL), Merck Healthcare KGaA, GLS (Univ. Waterloo), TÜV Holding GmbH, Software Competence Center Hagenberg GmbH, dSPACE GmbH, TRUMPF SE + Co. KG.

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

# A Derivations and Proofs

## A.1 Reverse KL Divergence with Reparametrization Trick

A popular choice for training diffusion bridges is the rKL objective [Richter and Berner, 2024, Vargas et al., 2024, Chen et al., 2024a, Blessing et al., 2024, He et al., 2025, Blessing et al., 2025] which is given by:

$$D_{KL}(q_{\alpha,\nu}(X_{0:T}) \| p_{\phi,\nu}(X_{0:T})) = \mathbb{E}_{X_{0:T} \sim q_{\alpha,\nu}(X_{0:T})} \left[ \log \frac{q_{\alpha,\nu}(X_{0:T})}{p_{\phi,\nu}(X_{0:T})} \right] \tag{14}$$

This loss involves an expectation over the variational distribution $q_{\alpha,\nu}$. This is a typical use case for the reparameterization trick [Glasserman, 2004]. This method gained popularity as a gradient estimator for training generative models [Kingma and Welling, 2014, Rezende et al., 2014]. This technique frequently yields a lower variance than the log-derivative trick (see Mohamed et al. [2020] for a discussion). In the context of diffusion samplers, the reparameterization trick was introduced in [Zhang and Chen, 2022]. In this method the trajectories of the reverse process $X_{0:T} \sim q_{\alpha,\nu}(X_{0:T})$ are reparameterized as a function $f_{\alpha,\nu}(\epsilon_{0:T}) := [g_{\alpha,\nu,0}, ..., g_{\alpha,\nu,T-1}]$ of independent Gaussian noise $\epsilon_t \sim \mathcal{N}(0, \mathrm{I})$ and the model parameters $(\alpha, \nu)$. Samples $X_t$ at time step $t$ are successively generated via the Euler-Maruyama update function $g_{\alpha,\nu,t} := X_{t-1} = g_{\alpha,\nu}(X_t, t, \epsilon_t)$, which is used to numerically integrate the reverse process Eq. 3. Substituting this reparameterization into the rKL loss yields, with slight abuse of notation:

$$D_{KL}(q_{\alpha,\nu}(\epsilon_{0:T}) \| p_{\phi,\nu}(\epsilon_{0:T})) = \mathbb{E}_{\epsilon_{0:T} \sim \mathcal{N}(0, I)} \left[ \log \frac{q_{\alpha,\nu}(f_{\alpha,\nu}(\epsilon_{0:T}))}{p_{\phi,\nu}(f_{\alpha,\nu}(\epsilon_{0:T}))} \right]. \tag{15}$$

This expression highlights that the gradients of the loss with respect to $(\alpha, \nu)$ can propagate into the expectation of the deterministic function $f_{\alpha,\nu}$, enabling direct gradient calculation. In diffusion samplers the frequent iterative application of $g_{\alpha,\nu,t}$ is likely to contribute to the vanishing or exploding gradient problem, which might explain the suboptimal behavior of rKL loss when minimized with usage of the reparametrization trick Zhang and Chen [2022]. Moreover, Vargas et al. [2023] demonstrates that applying a stop-gradient operation to the Langevin parametrization term in the PISgradNet architecture (see App. D.3) enhances the stability of the reparametrization trick in diffusion samplers. Without this modification, the direct application of the reparametrization trick through the gradient of the energy function is unstable, though the stop-gradient approach introduces an additional bias into the gradient.

## A.2 Definition of $\nabla_\theta^{LD,\omega_\theta}$ Operator

Let $\omega_\theta(X)$ be a variational probability distribution parameterized by $\theta$ and let $O_\theta(X)$ be a function that depends on these parameters. We want to define the $\nabla_\theta^{LD,\omega_\theta}$ operator based on the gradient of the expectation of the observable with respect to samples $X \sim \omega_\theta(X)$ given by $\langle O(X) \rangle_{X \sim \omega_\theta(X)}$.

The $\nabla_\theta^{LD,\omega_\theta}$ is then defined in the following way:

$$\nabla_\theta^{LD,q_\theta} \langle O_\theta(X) \rangle_{X \sim \omega_\theta(X)} := \mathbb{E}_{X \sim \omega_\theta(X)} \left[ \left( O_\theta(X) - b_\theta^{\omega_\theta} \right) \nabla_\theta \log \omega_\theta(X) \right] + \mathbb{E}_{X \sim \omega_\theta(X)} \left[ \nabla_\theta O_\theta(X) \right],$$

where $b_\theta^{\omega_\theta} = \mathbb{E}_{X \sim \omega_\theta} [\log \frac{\omega_\theta(X)}{p(X)}]$. When applying the $\nabla_\theta^{LD,\omega_\theta}$ operator we often use that in the case of $O(X) = \log \omega_\theta(X)$ we have $\mathbb{E}_{X \sim \omega_\theta(X)} [\nabla_\theta \log \omega_\theta(X)] = 0$.

## A.3 Gradient of the Log Variance Loss

In the following, we derive the gradient of the log variance loss, where we consider a reverse and forward diffusion process with shared parameters $\theta$:

$$\nabla_\theta D_{LV}^\omega(q_\theta, p) = \nabla_\theta \mathbb{E}_{X_{0:T}\sim\omega}\left[\left(\log\frac{q_\theta(X_{0:T})}{p_\theta(X_{0:T})} - \mathbb{E}_{X_{0:T}\sim\omega}\left[\log\frac{q_\theta(X_{0:T})}{p_\theta(X_{0:T})}\right]\right)^2\right]$$

$$= \mathbb{E}_{X_{0:T}\sim\omega}\left[2\left(\log\frac{q_\theta(X_{0:T})}{p_\theta(X_{0:T})} - \mathbb{E}_{X_{0:T}\sim\omega}\left[\log\frac{q_\theta(X_{0:T})}{p_\theta(X_{0:T})}\right]\right)\cdot\left(\nabla_\theta\log\frac{q_\theta(X_{0:T})}{p_\theta(X_{0:T})} - \mathbb{E}_{X_{0:T}\sim\omega}\left[\nabla_\theta\log\frac{q_\theta(X_{0:T})}{p_\theta(X_{0:T})}\right]\right)\right]$$

$$= \mathbb{E}_{X_{0:T}\sim\omega}\left[2\left(\log\frac{q_\theta(X_{0:T})}{p_\theta(X_{0:T})} - \mathbb{E}_{X_{0:T}\sim\omega}\left[\log\frac{q_\theta(X_{0:T})}{p_\theta(X_{0:T})}\right]\right)\cdot\nabla_\theta\log\frac{q_\theta(X_{0:T})}{p_\theta(X_{0:T})}\right]$$

Where we have used that

$$\mathbb{E}_{X_{0:T}\sim\omega}\left[2\left(\log\frac{q_\theta(X_{0:T})}{p_\theta(X_{0:T})} - \mathbb{E}_{X_{0:T}\sim\omega}\left[\log\frac{q_\theta(X_{0:T})}{p_\theta(X_{0:T})}\right]\right)\cdot\left(\mathbb{E}_{X_{0:T}\sim\omega}\left[\nabla_\theta\log\frac{q_\theta(X_{0:T})}{p_\theta(X_{0:T})}\right]\right)\right] = 0$$

since $\mathbb{E}_{X_{0:T}\sim\omega}\left[\nabla_\theta\log\frac{q_\theta(X_{0:T})}{p_\theta(X_{0:T})}\right]$ does not depend on $X_{0:T}$ and can be pulled out of the first expectation.

We can recover the gradients in Eq. 13 with respect to $\alpha$ by setting $\theta \to \alpha$, $q_\theta(X_{0:T}) \to q_\alpha(X_{0:T})$ and $p_\theta(X_{0:T}) \to p(X_{0:T})$. Likewise we can recover gradient with respect to $\phi$ with $q_\theta(X_{0:T}) \to q(X_{0:T})$ and $p_\theta(X_{0:T}) \to p_\phi(X_{0:T})$ and with respect to $\nu$ with $q_\theta(X_{0:T}) \to q_\nu(X_{0:T})$ and $p_\theta(X_{0:T}) \to p_\nu(X_{0:T})$.

### A.4 Gradient of the Reverse Kullback-Leibler Divergence Loss

In the following the gradient of the rKL divergence is derived, when it is optimized with the usage of the log-derivative trick:

$$\nabla_\theta D_{KL}(q_\theta(X_{0:T})\,\|\,p_\theta(X_{0:T})) = \nabla_\theta\mathbb{E}_{X_{0:T}\sim q_\theta}\left[\log\frac{q_\theta(X_{0:T})}{p_\theta(X_{0:T})}\right]$$

$$= \mathbb{E}_{X_{0:T}\sim q_\theta}\left[\log\frac{q_\theta(X_{0:T})}{p_\theta(X_{0:T})}\nabla_\theta\log q_\theta(X_{0:T})\right] + \mathbb{E}_{X_{0:T}\sim q_\theta}\left[\nabla_\theta\log\frac{q_\theta(X_{0:T})}{p_\theta(X_{0:T})}\right]$$

$$= \mathbb{E}_{X_{0:T}\sim q_\theta}\left[\left(\log\frac{q_\theta(X_{0:T})}{p_\theta(X_{0:T})} - \mathbb{E}_{X_{0:T}\sim q_\theta}\left[\log\frac{q_\theta(X_{0:T})}{p_\theta(X_{0:T})}\right]\right)\nabla_\theta\log q_\theta(X_{0:T})\right] + \mathbb{E}_{X_{0:T}\sim q_\theta}\left[\nabla_\theta\log\frac{q_\theta(X_{0:T})}{p_\theta(X_{0:T})}\right]$$

$$= \mathbb{E}_{X_{0:T}\sim q_\theta}\left[\left(\log\frac{q_\theta(X_{0:T})}{p_\theta(X_{0:T})} - \mathbb{E}_{X_{0:T}\sim q_\theta}\left[\log\frac{q_\theta(X_{0:T})}{p_\theta(X_{0:T})}\right]\right)\nabla_\theta\log q_\theta(X_{0:T})\right] - \mathbb{E}_{X_{0:T}\sim q_\theta}\left[\nabla_\theta\log p_\theta(X_{0:T})\right]$$

where we use in lines two to three the fact that $b\,\mathbb{E}_{X_{0:T}\sim q_\theta}\left[\nabla_\theta\log q_\theta(X_{0:T})\right] = 0$ and that $b = \mathbb{E}_{X_{0:T}\sim q_\theta}\left[\log\frac{q_\theta(X_{0:T})}{p_\theta(X_{0:T})}\right]$ is a baseline that leads to gradient updates with lower variance. In line three to four we have used that $\mathbb{E}_{X_{0:T}\sim q_\theta}\left[\nabla_\theta\log q_\theta(X_{0:T})\right] = \mathbb{E}_{X_{0:T}\sim q_\theta}\left[\frac{1}{q_\theta(X_{0:T})}\nabla_\theta q_\theta(X_{0:T})\right] = \int\nabla_\theta q_\theta(X_{0:T})d\,X_{0:T} = \nabla_\theta\int q_\theta(X_{0:T})\,dX_{0:T} = 0$

We can recover the gradients in Eq. 12 with respect to $\alpha$ by setting $\theta \to \alpha$, $q_\theta(X_{0:T}) \to q_\alpha(X_{0:T})$ and $p_\theta(X_{0:T}) \to p(X_{0:T})$. Likewise we can recover gradient with respect to $\phi$ with $q_\theta(X_{0:T}) \to q(X_{0:T})$ and $p_\theta(X_{0:T}) \to p_\phi(X_{0:T})$ and with respect to $\nu$ with $q_\theta(X_{0:T}) \to q_\nu(X_{0:T})$ and $p_\theta(X_{0:T}) \to p_\nu(X_{0:T})$.

### A.5 Counterexamples to Data Processing Inequality for the Log Variance Loss

#### A.5.1 Counterexample in the Continuous Domain

In the following we provide a counter example where

$$\text{Var}_{(x)\sim q}\left[\log\frac{q(x)}{p(x)}\right] \nleq \text{Var}_{(x,y)\sim q}\left[\log\frac{q(x,y)}{p(x,y)}\right] \tag{16}$$

where the distributions $q(x, y)$ is absolutely continuous with respect to $p(x, y)$ and the other way around. By definition, absolute continuity $q \ll p$ holds if every measurable set $A \subseteq [0, 1]^2$ for which $p(A) = 0$ also satisfies $q(A) = 0$.

**Consider the following distributions:**

$$p(x, y) = 1, \qquad q(x, y) = \begin{cases} a := \dfrac{2n}{n+1}, & x + y < 1, \\ b := \dfrac{2}{n+1}, & x + y \geq 1, \end{cases} \qquad (x, y) \in [0, 1]^2.$$

**Joint variance.** Since the lower and upper triangles $A = \{x + y < 1\}$ and $B = \{x + y \geq 1\}$ have equal area $\frac{1}{2}$, under $q$ the probabilities of the two constant values are $\Pr_q(A) = \frac{a}{2}$ and $\Pr_q(B) = \frac{b}{2}$. With $p(x, y) = 1$ we have $\log \frac{q(x,y)}{p(x,y)} = \log q(x, y)$ taking only values $\log a, \log b$. Hence

$$\mathbb{E}_q[\log q(X, Y)] = \frac{a}{2} \log a + \frac{b}{2} \log b,$$

$$\mathbb{E}_q[\log^2 q(X, Y)] = \frac{a}{2} \log^2 a + \frac{b}{2} \log^2 b,$$

and therefore

$$\boxed{\mathrm{Var}_{(X,Y) \sim q}\left[ \log \frac{q(X, Y)}{p(X, Y)} \right] = \frac{a}{2} \log^2 a + \frac{b}{2} \log^2 b - \left( \frac{a}{2} \log a + \frac{b}{2} \log b \right)^2.}$$

**Marginal variance.** First obtain the marginal $q(x)$:

$$q(x) = \int_0^1 q(x, y) \, dy = \int_0^{1-x} \frac{2n}{n+1} \, dy + \int_{1-x}^1 \frac{2}{n+1} \, dy = \frac{2}{n+1}\big(n + x(1 - n)\big).$$

Set

$$C := \frac{2}{n+1}, \qquad m := 1 - n, \qquad u := n + mx \quad (u \in [n, 1]).$$

Then, with the change of variables $u = n + mx$ (so $dx = \frac{du}{m}$), the moments under $X \sim q$ are

$$\mathbb{E}_q[\log q(X)] = \int_0^1 q(x) \log q(x) \, dx = \frac{C}{m} \int_n^1 u \log(Cu) \, du,$$

$$\mathbb{E}_q[\log^2 q(X)] = \frac{C}{m} \int_n^1 u \log^2(Cu) \, du.$$

Use $\log(Cu) = \log C + \log u$ and the elementary integrals

$$\int_n^1 u \, du = \frac{1 - n^2}{2},$$

$$\int_n^1 u \log u \, du = -\frac{1}{4} - \frac{n^2}{2} \log n + \frac{n^2}{4},$$

$$\int_n^1 u \log^2 u \, du = \frac{1}{4} - \frac{n^2}{2} \log^2 n + \frac{n^2}{2} \log n - \frac{n^2}{4}.$$

Let

$$A := \frac{1 - n^2}{2}, \quad B := -\frac{1}{4} - \frac{n^2}{2} \log n + \frac{n^2}{4}, \quad D := \frac{1}{4} - \frac{n^2}{2} \log^2 n + \frac{n^2}{2} \log n - \frac{n^2}{4}.$$

Then

$$\mathbb{E}_q[\log q(X)] = \frac{C}{m}\big((\log C)A + B\big),$$

$$\mathbb{E}_q[\log^2 q(X)] = \frac{C}{m}\big((\log C)^2 A + 2(\log C)B + D\big),$$

and hence

$$\boxed{\mathrm{Var}_{X \sim q}\left[ \log \frac{q(X)}{p(X)} \right] = \mathbb{E}_q[\log^2 q(X)] - \big(\mathbb{E}_q[\log q(X)]\big)^2,}$$

with the expectations given above (substitute $C = \frac{2}{n+1}$, $m = 1 - n$, and $A, B, D$ as defined).

**Numeric evaluation for** $n = 10^{-3}$. Substituting $n = 10^{-3}$ (so $a \approx 0.001998002$, $b \approx 1.998001998$) yields

$$\text{Var}_{(X,Y)\sim q}\left[\log\frac{q}{p}\right] \approx 0.04762179, \qquad \text{Var}_{X\sim q}\left[\log\frac{q}{p}\right] \approx 0.24995228 \approx 0.25,$$

showing the claimed DPI violation.

### A.5.2 Counterexample in the Discrete Domain

Let $p, q$ be probability distributions, similar as in Eq. 11 but without learnable parameters, with respective marginal and conditional distributions and $(X, Y) \sim q$. Then the data processing inequality for the LV loss is

$$\text{Var}_{(X)\sim q}\left[\log\frac{q(X)}{p(X)}\right] \leq \text{Var}_{(X,Y)\sim q}\left[\log\frac{q(X,Y)}{p(X,Y)}\right] \tag{17}$$

which can be decomposed such that

$$\text{Var}_{(X,Y)\sim q}\left[\log\frac{q(X,Y)}{p(X,Y)}\right] = \text{Var}_{(X,Y)\sim q}\left[\log\frac{q(X)}{p(X)}\right] + \text{Var}_{(X,Y)\sim q}\left[\log\frac{q(Y|X)}{p(Y|X)}\right]$$
$$+ 2\,\text{Cov}_{(X,Y)\sim q}\left[\log\frac{q(Y|X)}{p(Y|X)}, \log\frac{q(X)}{p(X)}\right].$$

Therefore, if we find distributions $p, q$ such that

$$\text{Var}_{(X,Y)\sim q}\left[\log\frac{q(Y|X)}{p(Y|X)}\right] + 2\,\text{Cov}_{(X,Y)\sim q}\left[\log\frac{q(Y|X)}{p(Y|X)}, \log\frac{q(X)}{p(X)}\right] \leq 0$$

we disprove inequality Eq. 17. For this, let $\mathcal{X} := \{0, 1\}$, $p, q$ be defined on $\mathcal{X} \times \mathcal{X} := \mathcal{X}^2$ with marginal probabilities

$$p(X = 0) = 0.1,\ p(X = 1) = 0.9, \quad q(X = 0) = 0.9,\ q(X = 1) = 0.1, \quad q(Y = 0) = 1,\ q(Y = 1) = 0$$

and conditional probabilities

$$p(Y = 0|X = 0) = 0.5,\ p(Y = 0|X = 1) = 0.1.$$

From $q(Y = 1) = 0$ we get $q(X, Y = 1) = 0$. By standard arguments e.g., as used for KL-divergence we can interpret

$$q(X, Y = 1)\log q(Y = 1|X) = q(X)q(Y = 1|X)\log q(Y = 1|X)$$

as being zero since $\lim_{x\to 0^+} x\log x = 0$ which results in the following simplifications

$$\text{Var}_{(X,Y)\sim q}\left[\log\frac{q(Y|X)}{p(Y|X)}\right] = \sum_{(x,y)\in\mathcal{X}^2} q(x,y)\left(\log\frac{q(y|x)}{p(y|x)} - \sum_{(x',y')\in\mathcal{X}^2} q(x',y')\log\frac{q(y'|x')}{p(y'|x')}\right)^2$$

$$= \sum_{x\in\mathcal{X}} q(x,0)\left(\log\frac{q(0|x)}{p(0|x)} - \sum_{x'\in\mathcal{X}} q(x',0)\log\frac{q(0|x')}{p(0|x')}\right)^2$$

$$= \sum_{x\in\mathcal{X}} q(x)\left(\log p(0|x) - \sum_{x'\in\mathcal{X}} q(x')\log p(0|x')\right)^2$$

by using $q(x, 0) = q(x)$. Analogously we get for

$$\text{Cov}_{(X,Y)\sim q}\left[\log\frac{q(Y|X)}{p(Y|X)}, \log\frac{q(X)}{p(X)}\right]$$

$$= -\sum_{x\in\mathcal{X}} q(x)\left(\log p(0|x) - \sum_{x'\in\mathcal{X}} q(x')\log p(0|x')\right)\left(\log\frac{q(x)}{p(x)} - \sum_{x'\in\mathcal{X}} q(x')\log\frac{q(x')}{p(x')}\right).$$

By inserting the corresponding probability values we have

$$\text{Cov}_{(X,Y)\sim q}\left[\log\frac{q(Y|X)}{p(Y|X)}, \log\frac{q_X(X)}{p_X(X)}\right] \approx -0.6365, \quad \text{Var}_{(X,Y)\sim q}\left[\log\frac{q(Y|X)}{p(Y|X)}\right] \approx 0.2331$$

which demonstrates that the data processing inequality does not always hold for the LV loss.

# B    Additional Experiments

We extend our experimental evaluation to include multimodal benchmarks in higher dimensions such as GMM-40 (100d) and MoS-10 (100d). The results in Table 4 and 5 demonstrate that the proposed rKL-LD loss achieves performance comparable to the LV loss with respect to Sinkhorn distance and Entropic Mode Coverage (EMC), while yielding superior results in terms of the Evidence Lower Bound (ELBO), log Z, and Maximum Mean Discrepancy (MMD). On GMM-100D, we observe that combining the LV loss with CMCD leads to divergent behavior in several random seed runs, substantially degrading the average performance metrics. A similar issue arises on MoS-100D when training DBS with the LV loss. Notably, learnable diffusion coefficients consistently enhance performance when trained with the rKL-LD loss, but degrade performance when used with the LV loss.

| Task | GMM-40 (100d) | | | | |
|---|---|---|---|---|---|
| Metric | Sinkhorn ($\downarrow$) | ELBO ($\uparrow$) | log Z ($\uparrow$) | EMC ($\uparrow$) | MMD ($\downarrow$) |
| **CMCD: rKL-R** $\star$ | $7152.88_{\pm177.312}$ | $-795.09_{\pm6.491}$ | $-528.09_{\pm2.995}$ | $0.995_{\pm0.000}$ | $0.116_{\pm0.001}$ |
| **CMCD: LV** $\star$ | $22237.03_{\pm10627.664}$ | $(-1.96_{\pm1.35}) \times 10^9$ | $-774.91_{\pm466.647}$ | $0.866_{\pm0.084}$ | $0.268_{\pm0.101}$ |
| **CMCD: LV** | $12420.83_{\pm6668.170}$ | $(-2.14_{\pm2.03}) \times 10^9$ | $-1037.09_{\pm931.118}$ | $0.951_{\pm0.043}$ | $0.138_{\pm0.015}$ |
| **CMCD: rKL-LD** $\star$ | $6079.38_{\pm205.248}$ | $-45.10_{\pm0.091}$ | $-18.09_{\pm0.341}$ | $\mathbf{0.996_{\pm0.000}}$ | $0.084_{\pm0.001}$ |
| **CMCD: rKL-LD** | $\mathbf{6043.26_{\pm166.427}}$ | $\mathbf{-45.84_{\pm0.080}}$ | $\mathbf{-18.88_{\pm0.394}}$ | $\mathbf{0.996_{\pm0.000}}$ | $0.085_{\pm0.001}$ |
| **DBS: rKL-R** $\star$ | $5637.36_{\pm155.953}$ | $-270.61_{\pm0.275}$ | $-138.60_{\pm0.850}$ | $\mathbf{0.997_{\pm0.001}}$ | $0.179_{\pm0.000}$ |
| **DBS: LV** $\star$ | $\mathbf{5024.58_{\pm171.079}}$ | $-230.83_{\pm0.151}$ | $-105.53_{\pm0.972}$ | $0.998_{\pm0.001}$ | $0.178_{\pm0.000}$ |
| **DBS: LV** | $4938.60_{\pm103.434}$ | $-252.56_{\pm0.299}$ | $-116.76_{\pm1.061}$ | $0.998_{\pm0.001}$ | $0.184_{\pm0.001}$ |
| **DBS: rKL-LD** $\star$ | $5163.31_{\pm206.480}$ | $-225.99_{\pm0.132}$ | $-101.80_{\pm0.953}$ | $0.997_{\pm0.001}$ | $0.178_{\pm0.000}$ |
| **DBS: rKL-LD** | $5132.65_{\pm124.068}$ | $\mathbf{-198.53_{\pm0.127}}$ | $\mathbf{-87.59_{\pm0.818}}$ | $0.997_{\pm0.001}$ | $\mathbf{0.171_{\pm0.000}}$ |

Table 4: Results on GMM-100D tasks. Arrow $\downarrow$ denotes that lower values of the metric are better, and $\uparrow$ denotes that higher values are better. $\star$ denotes that $\sigma_{\mathrm{diff}}$ is not learned during training. Best results are highlighted in bold.

| Task | MoS-10 (100d) | | | | |
|---|---|---|---|---|---|
| Metric | Sinkhorn ($\downarrow$) | ELBO ($\uparrow$) | log Z ($\uparrow$) | EMC ($\uparrow$) | MMD ($\downarrow$) |
| **CMCD: rKL-R** $\star$ | $1809.48_{\pm298.512}$ | $-114.86_{\pm0.021}$ | $-68.25_{\pm0.48}$ | $0.977_{\pm0.004}$ | $0.265_{\pm0.001}$ |
| **CMCD: LV** $\star$ | $2716.16_{\pm808.633}$ | $-97.44_{\pm0.622}$ | $-66.37_{\pm0.404}$ | $\mathbf{0.981_{\pm0.005}}$ | $0.286_{\pm0.001}$ |
| **CMCD: rKL-LD** $\star$ | $\mathbf{1852.13_{\pm271.333}}$ | $\mathbf{-68.21_{\pm0.322}}$ | $\mathbf{-31.71_{\pm0.173}}$ | $0.957_{\pm0.007}$ | $\mathbf{0.260_{\pm0.001}}$ |
| **DBS: rKL-R** $\star$ | $9732.62_{\pm3761.867}$ | $-183.53_{\pm0.867}$ | $-113.88_{\pm1.51}$ | $0.990_{\pm0.003}$ | $0.298_{\pm0.001}$ |
| **DBS: LV** $\star$ | $(1.96_{\pm0.78}) \times 10^{12}$ | $(-1.68_{\pm0.67}) \times 10^5$ | $-48409.64_{\pm18962.519}$ | $0.393_{\pm0.201}$ | $0.298_{\pm0.001}$ |
| **DBS: rKL-LD** $\star$ | $\mathbf{3626.16_{\pm1557.097}}$ | $\mathbf{-121.77_{\pm0.926}}$ | $\mathbf{-69.53_{\pm0.347}}$ | $\mathbf{0.996_{\pm0.001}}$ | $\mathbf{0.295_{\pm0.001}}$ |

Table 5: Results on MoS-100D tasks. Arrow $\downarrow$ denotes that lower values of the metric are better, and $\uparrow$ denotes that higher values are better. $\star$ denotes that $\sigma_{\mathrm{diff}}$ is not learned during training. Best results are highlighted in bold.

# C    Metrics

**Evidence Lower Bound**    The evidence lower bound is a lower bound on $\log \mathcal{Z}$ and is computed with:

$$\mathrm{ELBO} = \mathbb{E}_{X_{0:T} \sim q_{\alpha,\nu}(X_{0:T})} \left[ \frac{p_{\phi,\nu}(X_{0:T})}{q_{\alpha,\nu}(X_{0:T})} \right] \leq \log \mathcal{Z} \tag{18}$$

**Sinkhorn Distance**    The Sinkhorn distance is an entropic regularization of the 2-Wasserstein ($\mathcal{W}_2$) optimal transport (OT) distance Peyré et al. [2019], providing a principled alternative to ELBO for evaluating sample quality. Unlike ELBO, which is often insensitive to mode collapse Blessing et al. [2024], the Sinkhorn distance measures the discrepancy between generated and target distributions, offering better insights into sample diversity and multimodal coverage. As it requires access to ground-truth samples, its use is limited to synthetic tasks Chen et al. [2024a]. Following Blessing

et al. [2024], Chen et al. [2024a], we compute the Sinkhorn distance using the `ott` package Cuturi et al. [2022] and report it as a primary metric for the appropriate benchmarks. Sinkhorn distances are computed on Funnel using 2000 samples and on GMM and MoS using 16000 samples. On MayWell Sinkhorn distances are computed with 100000 samples using code from Richter and Berner [2024].

**Maximum Mean Discrepancy** Given samples $X \sim p(X)$ and samples $Y \sim q(Y)$ the Maximum Mean Discrepancy (MMD) is computed in the following way:

$$\text{MMD}(p,q) = \sqrt{\mathbb{E}_{X,X'\sim p(X)}\left[k(X,X')\right] + \mathbb{E}_{Y,Y'\sim q(Y)}\left[k(Y,Y')\right] - 2\mathbb{E}_{X\sim p(X),Y\sim q(Y)}\left[k(X,Y)\right]},$$

where $k(x,y) = \sum_{i=1}^{K} \exp\left(-\frac{1}{\kappa_i^2}\|x-y\|^2\right)$ where $\kappa_i$ is the bandwith. We follow He et al. [2025] and compute the MMD with an average bandwidth of $100$ and using $K = 10$ kernels. The code is based on an implementation of Chen et al. [2024b].

**Entropic Mode Coverage** The Entropic Mode Coverage Blessing et al. [2024] is given by

$$\text{EMC} := \mathbb{E}_{X_0\sim q_\theta}\left[\mathcal{H}(p(\xi, X_0))\right] = -\frac{1}{N}\sum_{X_0\sim q_\theta}\sum_{i=1}^{M} p(\xi, X_0)\log_M p(\xi, X_0)$$

where $i \in \{1, ..., M\}$ and $p(\xi_i, X_0)$ is the probability corresponding to the mixture component with the highest likelihood at $X_0$. The optimal value of EMC is $1$, i.e. every mode from the target distribution is covered and it is $0$. in the worst case.

## C.1 Benchmarks

### C.1.1 Bayesian Learning tasks

These tasks involve probabilistic inference where the true underlying parameter distributions are unknown, requiring Bayesian approaches for estimation.

**Bayesian Logistic Regression (Sonar and Credit).** We consider Bayesian logistic regression for binary classification on two well-established benchmark datasets, frequently used for evaluating variational inference and Markov Chain Monte Carlo (MCMC) methods. The model's posterior distribution is given by:

$$\rho_{\text{target}}(x) = p(x)\prod_{i=1}^{n} \text{Bernoulli}\left(y_i; \text{sigmoid}(x \cdot u_i)\right)$$

where the dataset consists of standardized input-output pairs $((u_i, y_i))_{i=1}^{n}$. Our evaluation includes the Sonar dataset ($d = 61, n = 208$) and the German Credit dataset ($d = 25, n = 1000$). The prior distribution is chosen as a standard Gaussian $p = \mathcal{N}(0, I)$ for Sonar, whereas for German Credit, we follow the implementation of Blessing et al. [2024], which omits an explicit prior by setting $p \equiv 1$.

**Random Effect Regression (Seeds):** The Seeds dataset ($d = 26$) is modeled using a hierarchical random effects regression framework, which captures both fixed and random effects to account for variability across different experimental conditions. The generative process is specified as:

$$\tau \sim \text{Gamma}(0.01, 0.01)$$
$$a_0, a_1, a_2, a_{12} \sim \mathcal{N}(0, 10)$$
$$b_i \sim \mathcal{N}\left(0, \frac{1}{\sqrt{\tau}}\right), \quad i = 1, \dots, 21,$$
$$\text{logits}_i = a_0 + a_1 x_i + a_2 y_i + a_{12} x_i y_i + b_1, \quad i = 1, \dots, 21,$$
$$r_i \sim \text{Binomial}\left(\text{logits}_i, N_i\right), \quad i = 1, \dots, 21.$$

The inference task involves estimating the posterior distributions of $\tau$, $a_0$, $a_1$, $a_2$, $a_{12}$, and the random effects $b_i$, given observed values of $x_i$, $y_i$, and $N_i$. This model is particularly relevant for analyzing seed germination proportions, where the inclusion of random effects accounts for heterogeneity in experimental conditions; see Geffner and Domke [2023] for further details.

**Time Series Models (Brownian):** The Brownian motion model ($d = 32$) represents a discretized stochastic process commonly used in time series analysis, with Gaussian observation noise. The generative model follows:

$$\begin{aligned}
\alpha_{\text{inn}} &\sim \text{LogNormal}(0, 2), \\
\alpha_{\text{obs}} &\sim \text{LogNormal}(0, 2), \\
x_1 &\sim \mathcal{N}(0, \alpha_{\text{inn}}), \\
x_i &\sim \mathcal{N}(x_{i-1}, \alpha_{\text{inn}}), \quad i = 2, \dots, 30, \\
y_i &\sim \mathcal{N}(x_i, \alpha_{\text{obs}}), \quad i = 1, \dots, 30.
\end{aligned}$$

The inference objective is to estimate $\alpha_{\text{inn}}$, $\alpha_{\text{obs}}$, and the latent states $\{x_i\}_{i=1}^{30}$ based on the available observations $\{y_i\}_{i=1}^{10}$ and $\{y_i\}_{i=20}^{30}$, with the middle observations missing. This missing-data structure increases the difficulty of inference, making it a useful benchmark for probabilistic time series modeling; see Geffner and Domke [2023].

**Spatial Statistics (LGCP):** The *Log-Gaussian Cox Process* (LGCP) is a widely used spatial model in statistics [Møller et al., 1998], which describes spatially distributed point processes such as the locations of tree saplings. The target density is defined over a discretized spatial grid of size $d = 40 \times 40 = 1600$, and follows:

$$\rho_{\text{target}} = \mathcal{N}(x; \mu, \Sigma) \prod_{i=1}^{d} \exp\left(x_i y_i - \frac{\exp(x_i)}{d}\right),$$

where $y$ represents an observed dataset, and $\mu$ and $\Sigma$ define the mean and covariance of the prior distribution. This formulation leads to a complex spatial dependency structure. We focus on the more challenging *unwhitened* variant of the model, which retains the full covariance structure and thus introduces stronger dependencies between grid locations, as described in Heng et al. [2020], Arbel et al. [2021].

### C.1.2 Synthetic targets:

For these tasks, ground-truth samples are available, allowing for direct evaluation of inference accuracy.

**Mixture distributions (GMM and MoS):** We consider mixture models where the target distribution consists of $m$ mixture components, defined as:

$$p_{\text{target}} = \frac{1}{m} \sum_{i=1}^{m} p_i.$$

The *Gaussian Mixture Model* (GMM), adapted from Blessing et al. [2024], is constructed with $m = 40$ Gaussian components:

$$\begin{aligned}
p_i &= \mathcal{N}(\mu_i, I), \\
\mu_i &\sim \mathcal{U}_d(-40, 40),
\end{aligned}$$

where $\mathcal{U}_d(l, u)$ denotes a uniform distribution over $[l, u]^d$. We set the dimensionality to $d = 50$ in the experiments in Tab. 2.

The *Mixture of Student's t-distributions* (MoS) follows a similar construction but uses Student's $t$-distributions with two degrees of freedom ($t_2$) as the mixture components:

$$\begin{aligned}
p_i &= t_2 + \mu_i, \\
\mu_i &\sim \mathcal{U}_d(-10, 10),
\end{aligned}$$

where $\mu_i$ denotes the translation of each component. We set the dimensionality to $d = 50$ in the experiments in Tab. 2.

For both the GMM and MoS tasks, the component locations $\mu_i$ remain fixed across experiments using a predefined random seed to ensure reproducibility.

**Funnel:** The *Funnel* distribution, originally introduced in Neal [2003], serves as a challenging benchmark due to its highly anisotropic shape. It is defined as:

$$p_{\text{target}}(x) = \mathcal{N}(x_1; 0, \sigma^2)\mathcal{N}(x_2, \ldots, x_{10}; 0, \exp(x_1)I), \tag{19}$$

where $\sigma^2 = 9$ for any number of dimensions $d \geq 2$. In our main experiments, we consider the case $d = 10$. To maintain consistency with prior benchmarks Blessing et al. [2024], we apply a hard constraint by clipping all sampled values to the interval $[-30, 30]$.

**Many Well:** A standard problem in molecular dynamics is sampling from the stationary distribution of a Langevin dynamic. For our experiments, the resulting $d$-dimensional *many-well* potential corresponds to the following unnormalized density

$$\rho_{\text{target}}(x) = \exp\left(-\sum_{i=1}^{m}(x_i^2 - \delta)^2 - \frac{1}{2}\sum_{i=m+1}^{d} x_i^2\right)$$

with separation parameter $\delta \in (0, \infty)$ and $m \in \mathbb{N}$ combined double wells. Following Chen et al. [2024a], Berner et al. [2022] we set $d = m = 5$ and $\delta = 4$ which results in $2^m = 32$ well-separated modes. Since $\rho_{\text{target}}$ factorizes in the dimensions ground truth statistics and samples can be obtained by numerical integration and rejection sampling.

## D  Experimental Details

### D.1  Evaluation

In the Bayesian learning task, we compute the average of the ELBO over three independent runs, each estimated using 2000 samples. For the LGCP task, the evaluation is performed using 300 samples. The ELBO values reported in Tab. 1 represent the best ELBO achieved during training. For synthetic tasks in Tab. 2 and Tab. 3 we report all metrics based on the checkpoint at the end of training. When the run diverges on synthetic benchmarks, we report the result based on the checkpoint with the best ELBO. We use 16000 samples on MoS-40 50D and GMM-40 50D, 2000 samples on Funnel, and 100000 samples on Many Well to estimate the Sinkhorn distance. In Tab. 1, Tab. 2 and Tab. 3 we report the average metric value together with the standard error over three seeds. For MoS-40 50D and GMM-40 50D we calculate the standard error over 7 independent seeds, where for each seed the metrics are averaged over 30 repetitions due to large standard errors of the sinkhorn distance. MMD values are computed by using 4000 samples. Ground truth Sinkhorn distances and MMD values are computed by calculating these distances between two independent set of samples from the target distribution. The ground truth values are then averaged over three independent seeds.

### D.2  Hyperparameter tuning

**Benchmarks** In benchmark experiments in Sec. 5 we perform for each method a grid search over $\sigma_{\text{diff}}$, $\sigma_{\text{prior}}$, the learning rate of the model. The learning rate of the diffusion parameters such as $\sigma_{\text{prior}}$ and $\sigma_{\text{diff}}$ is always chosen to be equal to the model learning rate. On all Bayesian learning tasks, we perform for CMCD and DBS a grid search over $\sigma_{\text{diff,init}} = \{0.1, 0.3\}$, $\sigma_{\text{prior,init}} = \{0.5, 1.0\}$ and the learning rate $\lambda_{\text{model,SDE}} \in \{0.005, 0.002, 0.001\}$. On Brownian and German Credit, we found that if $\sigma_{\text{diff}}$ is not learned a finer grid-search over $\sigma_{\text{diff}}$ is necessary. Therefore on Brownian, we additionally add $\sigma_{\text{diff}} = 0.05$ and on German Credit $\sigma_{\text{diff}} = 0.01$ to the grid search.

On MoS 50D and GMM 50D, we follow Chen et al. [2024a] and fix $\sigma_{\text{prior,init}}$ to a high initial value. We found that $\sigma_{\text{prior,init}} = 80$ yielded the best results. We found that small model learning rates and compared to that large learning rates of the interpolation parameters between the prior and the target distribution work well. Therefore we adapt the grid search to $\sigma_{\text{diff,init}} = \{1., 1.5\}$, $\lambda_{\text{interpol}} = \{0.01, 0.001\}$ and the learning rate $\lambda_{\text{model,SDE}} \in \{0.0001, 0.00005, 0.00001\}$ for CMCD. For DBS, the interpolation between the prior and the target distribution is not learned. Therefore, we did not additionally search over $\lambda_{\text{interpol}}$ but increased the size of the grid search by searching over $\sigma_{\text{prior,init}} = \{60, 80\}$. On Many Well we conduct grid search over $\sigma_{\text{diff,init}} = \{0.05, 0.1, 0.2\}$, $\sigma_{\text{prior,init}} = \{0.5, 1.0, 2.0\}$, $\lambda_{\text{model,SDE}} \in \{0.001, 0.0001, 0.00001\}$.

Grid searches are performed over 8000 training iterations on all targets except MoS 50d and GMM 50d, where 12000 training iterations are performed. The best run is chosen according to the best ELBO value at the end of training on Bayesian tasks and on Synthetic targets according to the run with the best Sinkhorn distance at the end of training. The best hyperparameters are then run for 40000 training iterations. On MoS 50d and GMM 50d in Tab. 2, the best Sinkhorn distance is sometimes achieved at initialization. In this case, the checkpoint is excluded as it has only slightly better Sinkhorn values but much worse ELBOs than the runs at the end of training.

All scripts for running grid searches can be found in the code in /Configs/Sweeps/, and the final selected hyperparameters can be found in /Configs/Sweeps/BestRunsSweeps/.

**Ablations**   In ablation experiments in Sec. 5 we iteratively tuned hyperparameters such as $\sigma_{\mathrm{prior,init}}$ and $\sigma_{\mathrm{diff,init}}$ and learning rates for each method. For CMCD-LV and CMCD-LV $\star$ we found it hard to find a good-performing diffusion coefficient. Therefore, we used the learned average diffusion coefficient of CMCD-rKL-LD $\sigma_{\mathrm{diff}}$ as a starting point for iterative hyperparameter tuning which resulted in a decent performance of CMCD-LV and CMCD-LV $\star$.

### D.3   Architecture

**Score parametrization**   The learned score with parameters $\theta$ is parameterized in the following way:

$$s_\theta(X_t) = \mathrm{clip}\big(\tilde{s}_\theta(X_t, t) + \hat{s}_\theta(t) \odot \mathrm{clip}(\nabla_{X_t} \log \pi_t(X_t), -10^2, 10^2), -10^4, 10^4\big) \quad (20)$$

where $\tilde{s}_\theta(X_t, t)$ and $\hat{s}_\theta(t)$ are two separate neural networks which are parameterized with the PIS-gradnet architecture from Vargas et al. [2024] with 64 hidden neurons and 2 layers. The score is related to the control by $u_\theta(X_t) = \sigma_t s_\theta(X_t)$. The usage of the $\nabla_{X_t} \log \pi_t(X_t)$ term in the neural network parametrization is often refered to as langevin parametrization [He et al., 2025].

**Parametrization of prior distribution:**   Similarly to Chen et al. [2024a] and Blessing et al. [2024] and parameterize the prior distribution $\pi_T$ in the following way:

$$\pi_T = \mathcal{N}(\mu_\theta, \mathrm{diag}(\exp(l_\theta)))$$

where $\mu_\theta \in \mathbb{R}^d$ and logarithmic standard deviations $l_\theta \in \mathbb{R}^d$ are learnable parameters. In contrast to Chen et al. [2024a] and Blessing et al. [2024] we do not update $\mu_\theta$ and $l_\theta$ via the reparameterization trick as training progresses but also with the usage of the log-derivative trick.

**Parametrization of interpolation parameters:**   For each diffusion time step $t \in \{0, ..., T-1\}$ we parameterize the interpolation parameter $\beta_\theta(t)$ in the following way:

$$\beta_\theta(t) = \frac{\mathrm{softplus}(\theta_t)}{\sum_{t=0}^{T} \mathrm{softplus}(\theta_t)}, \quad (21)$$

where $\theta \in \mathbb{R}^T$ are learnable parameters. Each variable of $\theta \in \mathbb{R}^T$ is initialized to zero.

**Parametrization diffusion coefficient:**   We keep diffusion coefficients constant across time steps and parameterize it as $\sigma_t = \exp \gamma$, where $\gamma = \log \sigma_{\mathrm{init}}$. In principle, one could parameterize it similarly as the interpolation parameters, which would allow for a time-adaptive schedule. However, we leave this for future work.

**Training**   All parameters are trained with the usage of the RAdam Liu et al. [2020] optimizer. We use gradient clipping by norm at the value of 1. The learning rate decays with a cosine learning rate schedule from $\lambda_{\mathrm{start}}$ to $\lambda_{\mathrm{start}}/10$.

### D.4   Pseudoalgorithms

In the following we provide pseudoalgorithms for the computation of the LV loss, rKL-LD loss and the rKL-R Loss:

**Algorithm 1** Computation of the LV Loss
___
1: **Given:** Batch of diffusion paths $X_{0:T} \sim q_{\alpha,\nu}$ computed with the Euler-Maruyama integration (Eq. 4)
2: $X'_{0:T} \leftarrow \text{stop\_grad}(X_{0:T})$ $\quad\quad\quad$ ▷ detach the gradient from the Euler-Maruyama integration
3: compute $\log q_{\alpha,\nu}(X'_{0:T})$ with Eq. 5
4: compute $\log p_{\phi,\nu}(X'_{0:T})$ with Eq. 6
5: compute Loss $L(\alpha,\phi,\nu) = Var\left(\log \frac{q_{\alpha,\nu}(X'_{0:T})}{p_{\phi,\nu}(X'_{0:T})}\right)$
6: Backpropagate through the loss and update $(\alpha,\phi,\nu)$ using Adam optimizer
___

**Algorithm 2** Computation of the rKL-R Loss
___
1: **Given:** Batch of diffusion paths $X_{0:T} \sim q_{\alpha,\nu}$ computed with the Euler-Maruyama integration (Eq. 4)
2: compute $\log q_{\alpha,\nu}(X_{0:T})$ with Eq. 5
3: compute $\log p_{\phi,\nu}(X_{0:T})$ with Eq. 6
4: compute Loss $L(\alpha,\phi,\nu) = mean\left[\log \frac{q_{\alpha,\nu}(X_{0:T})}{p_{\phi,\nu}(X_{0:T})}\right]$
5: Backpropagate through the loss and update $(\alpha,\phi,\nu)$ using Adam optimizer
___

**Algorithm 3** Computation of the rKL-LD Loss:
Averages are always computed over the batch dimension
___
1: **Given:** Batch of diffusion paths $X_{0:T} \sim q_{\alpha,\nu}(X_{0:T})$ computed with the Euler-Maruyama integration (Eq. 4)
2: $X'_{0:T} \leftarrow \text{stop\_grad}(X_{0:T})$
3: compute $\log q_{\alpha,\nu}(X'_{0:T})$ with Eq. 5
4: compute $\log p_{\phi,\nu}(X'_{0:T})$ with Eq. 6
5: compute control variate $b^{q_{\alpha,\nu}}_{\alpha,\phi,\nu} = mean\left[\log \frac{q_{\alpha,\nu}(X'_{0:T})}{p_{\phi,\nu}(X'_{0:T})}\right]$
6: compute advantages $A^* = \text{stop\_gradient}\left[\log \frac{q_{\alpha,\nu}(X'_{0:T})}{p_{\phi,\nu}(X'_{0:T})} - b^{q_{\alpha,\nu}}_{\alpha,\phi,\nu} \vec{1}\right]$
7: compute Loss $L(\alpha,\phi,\nu) = mean\left[A^* \odot \log q_{\alpha,\nu}(X'_{0:T})\right] - mean\left[\log p_{\phi,\nu}(X'_{0:T})\right]$
8: Backpropagate through the loss and update $(\alpha,\phi,\nu)$ using Adam optimizer
___

