# OpenReview forum: "Rethinking Losses for Diffusion Bridge Samplers"
_NeurIPS.cc/2025/Conference — NeurIPS 2025 poster_

### Official Review · Reviewer_iCJN · 2025-06-11

**Clarity:** 3
**Significance:** 3
**Originality:** 2
**Rating:** 4
**Confidence:** 2

**Summary:**

This paper reveals the relationship between the gradients of Log Variance(LV) loss and the reverse Kullback-Leibler (rKL) loss for various cases. It also observes that for diffusion bridges, the LV loss does not satisfy the data processing inequality. Therefore, the authors argue that LV loss is not a good choice for diffusion bridges and advocate for training diffusion bridge samplers with the rKL loss with the log-derivative trick.
Experiments show that it consistently outperforms diffusion samplers trained with LV and original rKL-R losses.

**Questions:**

1. Larger datasets should also be verified.
2. Clarify the problems written above about the data processing inequality and the advantage of this method.

**Ethical Concerns:**

["NO or VERY MINOR ethics concerns only"]

**Final Justification:**

Since most of my concerns have been addressed, I raise my score to 4.

**Limitations:**

Yes

**Quality:**

2

**Strengths And Weaknesses:**

Strengths:
1. This paper gives a strong theoretical analysis of the relationship between the gradients of Log Variance(LV) loss and the reverse Kullback-Leibler (rKL) loss, and finds that their gradients do not hold in the context of diffusion bridge samplers or when diffusion coefficients are learned. This analysis provides more insights for understanding these two common losses.

2. This paper demonstrates that using rKL loss with the log-derivative trick can achieve better performance and is significantly less sensitive to hyperparameter tuning. Combined with learnable diffusion terms, the performance can be improved further.

Weakness:
1. The paper shows a counter-example to prove LV loss does not satisfy the data processing inequality, but for their counter-example, the distribution p(X, Y) is a discrete distribution and p(Y) is a $\delta$-distribution, which seems not rational for its assumption, Y is a diffused distribution from X. So I don't think this counter-example is convincing to support their points.

2. Just for my understanding, no matter that the LV loss does not satisfy the data processing inequality or the non-equivalence gradients between LV loss and rKL loss, they can't strongly support that the LV loss is not better than rKL loss from a theoretical view.  Theoretically, at least LV loss can converge to the optimal only when the two diffusion processes are identical. Therefore, the advantage of the proposed method is mostly from experimental observation rather than a convincing theoretical analysis.
3. For experiments, the proposed method is only verified in some small-dimensional datasets. Can it also be verified on a larger dataset(larger than 1000 or 10000)?

---

> ### Author Rebuttal · Authors · 2025-07-31
>
> We appreciate the reviewer's thoughtful feedback and are grateful for the acknowledgment of our "strong theoretical analysis" regarding the relationship between the gradients of the Log Variance (LV) loss and the reverse KL divergence (rKL) loss. We are also pleased that our experiments demonstrate the consistent superiority of our proposed loss over the LV and rKL-R losses.
>
> **Response to Comment W1: Counterexample is not convincing**
>
> For diffusion bridge samplers in the discrete domain, the learnable prior distribution $ q(Y) $ could indeed becomes a $ \delta $ distribution. So we believe that this example is a valid counterexample.
> However, to further illustrate our point of the DPI violation, we provide an additional counterexample based on continuous distributions:
>
> Consider the following distributions:
> - $ q(x, y) $ is uniform on the triangle defined by the vertices $ (0,0) $, $ (1,0) $, and $ (0,1) $.
> - $ p(x, y) $ is uniform on the entire unit square $ [0,1]^2 $.
> Thus, we have:
> - $ q(x) = 2  (1 - x) $
> - $ p(x) = 1 $
>
> In this example $q(y|x) = 1/(1-x)$ for  $y \in [0,1-x]$ and $0$ otherwise. Thus $y$ does contain information on $x$, like in a diffusion process.
>
> This example yields:
> - $ \mathrm{var}_{x,y \sim q(x,y)} \log \frac{q(x,y)}{p(x,y)} = 0 $ for the joint distributions.
> - $ \mathrm{var}_{x \sim q(x)} \left( \log \frac{q(x)}{p(x)} \right) \approx 0.25$ for the marginals $ q(x) $ and $ p(x) $.
>
> Hence, this simple example again demonstrates that the LV violates the DPI. The distribution $ q(x, y) $ can be regarded as a Markov process and is thus relatable to discretized diffusion processes on continuous domains. Please also note that in this example, the LV for the joints vanishes, i.e., it is at a global minimum. However, the marginals are still different, which shows that the LV loss can be a misleading loss function for the goal of approximating the target distribution.
> We will include this example in the appendix.
>
> **Response to Comment W2: Theoretical convergence of LV loss**
>
> The statement that "Theoretically, at least LV loss can converge to the optimal only when the two diffusion processes are identical" is not necessarily true. This holds only when the support of the proposal distribution includes the support of the variational and the target distribution, which cannot be guaranteed during training. Our newly provided counterexample shows that the LV loss of the joint distributions can be zero even though $ p(x, y) $ and $ q(x, y) $ are not identical. This is particularly relevant for the On-Policy LV loss (OP-LV), where the assumption is certainly not satisfied, and thus the OP-LV loss might possess suboptimal solutions as global minima. For other divergences commonly known in information geometry, such failure cases do not exist.
>
> **Response to Comment W3: Verification on larger datasets**
>
> We would like to emphasize that our experiments on the Log-Gaussian Cox Process (LGCP) involve 1600-dimensional data, where we observe that the rKL-LD loss outperforms the LV loss in all four cases. To our knowledge, there are no other popular benchmarks in dimensions of 1000 or 10000 in recent literature (see, e.g., [1], [2], [3]).
> We would like to highlight the comprehensiveness of our experimental evaluation. Specifically, we tested 5 different loss functions across 5 Bayesian learning tasks and 4 multimodal benchmarks, each assessed using 2 distinct diffusion bridge samplers. For each configuration, we performed a grid search followed by final run evaluations. In total, this resulted in over 1,350 individual experiments (calculated as 5 × 9 × 2 × 12 + 5 × 9 × 2 × 3). This level of experimental rigor aligns with the standard protocol commonly adopted in published work within this domain [2].
> Evaluating multimodal distributions in such high dimensions is memory-intensive. Computing Maximum Mean Discrepancy (MMD) and Sinkhorn distances requires many samples to achieve sufficient precision, which incurs higher memory costs. Our current code runs out of memory, and we would need more time and several code changes to evaluate MMDs and Sinkhorn distances in such high dimensions.
> Nevertheless, we have conducted experiments on multimodal target distributions in higher dimensions using GMM-100D and MoS-100D (see below). Our results indicate that the rKL-LD loss performs comparably to the LV loss in terms of Sinkhorn distance and Entropic Mode Coverage (EMC), but outperforms the LV loss in terms of Evidence Lower Bound (ELBO), log Z, and Maximum Mean Discrepancy (MMD). On GMM-100D, the LV loss combined with CMCD exhibits divergent behavior in a few seed runs, which significantly degrades its average performance metrics. This problem also occured in MoS-100D for DBS trained with the LV loss. Notably, we again observe that learnable diffusion coefficients improve performance when trained with rKL-LD but worsen performance when trained with LV.
>
> **Response to Limitations: Clarify the advantage of this method.**
>
> As stated in several sections of our paper and as the reviewer has stated in the “Summary” and “Strength” sections of their review, our proposed loss offers multiple advantages. It is less sensitive to hyperparameters, more stable during training, and achieves superior results across a wide range of benchmarks. Unlike the LV loss, our method remains stable when diffusion coefficients are trained, and its performance can be further enhanced by learning these coefficients.
> We thank you once again for the time and effort you have dedicated to reviewing our work. We hope that our responses have adequately addressed your questions.
>
>
> [1] Sendera, Marcin, et al. "Improved off-policy training of diffusion samplers." Advances in Neural Information Processing Systems 37 (2024): 81016-81045.
>
> [2] Chen, Junhua, et al. "Sequential Controlled Langevin Diffusions." The Thirteenth International Conference on Learning Representations.
>
> [3] Blessing, Denis, et al. "Underdamped Diffusion Bridges with Applications to Sampling." The Thirteenth International Conference on Learning Representations.
>
>
> | GMM 100-D | Sinkhorn | ELBO | log Z | EMC | MMD |
> |----------|----------|----------|----------|----------|----------|
> | CMCD: rKL-R $\Large \star$ | $7152.88\pm 177.312$    | $-795.09\pm 6.491$    | $-528.09\pm 2.995$     | $0.995\pm 0.000$     | $0.116\pm 0.001$     |
> | CMCD: LV $\Large \star$   | $22237.03\pm 10627.664$    | $-1957894528.00\pm 1354142383.491$    | $-774.91\pm 466.647$   | $0.866\pm 0.084$    | $0.268\pm 0.101$    |
> | CMCD: LV  | $12420.83\pm 6668.17$   | $-2141430400.00\pm 2031539132.986$    | $-1037.09\pm 931.118$    | $0.951\pm 0.043$    | $0.138\pm 0.015$    |
> | CMCD: rKL-LD $\Large \star$   | $6079.38\pm 205.248$     | $-45.10 \pm 0.091$    | $-18.09\pm 0.341$    |  $\mathbf{0.996 \pm 0.000}$   | $0.084\pm 0.001$    |
> | CMCD: rKL-LD    | $\mathbf{6043.26\pm 166.427}$     | $\mathbf{-45.84\pm 0.08}$    | $\mathbf{-18.88\pm 0.394}$   | $\mathbf{0.996\pm 0.000}$     | $\mathbf{0.085\pm 0.001}$    |
>
> | GMM 100-D | Sinkhorn | ELBO | log Z | EMC | MMD |
> |----------|----------|----------|----------|----------|----------|
> | DBS: rKL-R $\Large \star$    | $5637.36 \pm 155.953$     | $-270.61 \pm 0.275$    | $-138.60\pm 0.85$    | $\mathbf{0.997\pm 0.001}$ | $0.179\pm 0.000$   |
> | DBS: LV $\Large \star$  | $\mathbf{5024.58 \pm 171.079}$     | $-230.83\pm 0.151$    |$-105.53\pm 0.972$    | $\mathbf{0.998\pm 0.001}$    | $0.178\pm 0.000$    |
> | DBS: LV   | $\mathbf{4938.60\pm 103.434}$   | $-252.56\pm 0.299$    | $-116.76\pm 1.061$   | $\mathbf{0.998\pm 0.001}$    |  $0.184\pm 0.001$    |
> | DBS: rKL-LD $\Large \star$   | $\mathbf{5163.31\pm 206.48}$     | $-225.99 \pm 0.132$     | $-101.80\pm 0.953$    | $\mathbf{0.997\pm 0.001}$     | $0.178 \pm 0.000$    |
> | DBS: rKL-LD   | $\mathbf{5132.65\pm 124.068}$   | $\mathbf{-198.53\pm 0.127}$    | $\mathbf{-87.59\pm 0.818}$    | $\mathbf{0.997\pm 0.001}$     | $\mathbf{0.171\pm 0.000}$     |
>
>
> | MoS 100 D | Sinkhorn | ELBO | log Z | EMC | MMD |
> |----------|----------|----------|----------|----------|----------|
> | CMCD: rKL-R $\Large \star$ | $1809.48\pm 298.512$	| $-114.86\pm 0.021$	| $-68.25\pm 0.48$ 	| $0.977\pm 0.004$ 	| $0.265\pm 0.001$ 	|
> | CMCD: LV $\Large \star$	| $2716.16\pm 808.633$	| $-97.44\pm 0.622$	| $-66.37\pm 0.404$   | $\mathbf{0.981\pm 0.005}$	| $0.286\pm 0.001$	|
> | CMCD: rKL-LD $\Large \star$  | $\mathbf{1852.13 \pm 271.333}$ 	| $\mathbf{-68.21\pm 0.322}$	| $\mathbf{-31.71\pm 0.173}$	|  $0.957\pm 0.007$   |$\mathbf{0.260\pm 0.001}$	|
>
> | MoS 100 D | Sinkhorn | ELBO | log Z | EMC | MMD |
> |----------|----------|----------|----------|----------|----------|
> | DBS: rKL-R $\Large \star$ | $9732.62\pm 3761.867$	| $-183.53\pm 0.867$	| $-113.88\pm 1.51$ 	| $0.990\pm 0.003$ 	| $0.298\pm 0.001$ 	|
> | DBS: LV $\Large \star$  | $1964053495808.00 \pm 783156518610.415$ 	| $-168435.12 \pm 66865.384 $	|$-48409.64 \pm 18962.519 $	| $0.393\pm 0.201$	| $0.298\pm 0.001$	|
> | DBS: rKL-LD $\Large \star$   | $\mathbf{3626.16\pm 1557.097}$ 	| $\mathbf{-121.77 \pm 0.926}$ 	| $\mathbf{-69.53 \pm 0.347}$	| $\mathbf{0.996 \pm 0.001}$ 	| $\mathbf{0.295 \pm 0.001}$	|

---

> > ### Comment · Reviewer_iCJN · 2025-08-04
> >
> > Since most of my concerns have been addressed, including the counterexample and the convergence of LV loss, I raise my score to 4.

---

### Official Review · Reviewer_KBTa · 2025-06-30

**Clarity:** 2
**Significance:** 2
**Originality:** 3
**Rating:** 4
**Confidence:** 4

**Summary:**

This paper advocates for training diffusion samplers with the reverse Kullback-Leibler (rKL) loss using the log-derivative trick (rKL-LD), arguing it is more effective than the commonly used Log Variance (LV) loss. The authors demonstrate that the equivalence between LV and rKL losses is applicable with non-learnable forward processes, while it does not apply to diffusion bridges, and the LV loss fails to satisfy the data processing inequality in this context. Experimental results show that the rKL-LD loss consistently outperforms the LV loss on several benchmarks, offering more stable training and reduced sensitivity to hyperparameters, especially when diffusion coefficients are learned. While rKL-LD shows superior performance, the paper notes its potential for mode collapse if not well-tuned and suggests future work could explore its combination with off-policy sample buffers.

**Questions:**

When comparing the performance of rKL-LD to the LV loss, did your implementation of the LV baseline include a replay buffer? Recent work has shown that off-policy sample buffers significantly improve the stability and performance of LV-based training, making this a critical detail for a fair comparison.

**Ethical Concerns:**

["NO or VERY MINOR ethics concerns only"]

**Final Justification:**

I increased my score from 3 to 4 since most of my concerns have been addressed, such as the clarity in final objectives and the gradient equivalence. Although there still exists the limitation of empirical study, I think it is okay due to the scalability of the diffusion bridge sampler. I believe this paper still introduces good insights to re-understanding the connection between log-variance and KL-divergence.

**Limitations:**

The primary limitation is a potential contradiction in the paper's core theoretical argument, as the claimed gradient equivalence between rKL-LD and LV loss does not actually hold for practical implementations like CMCD where parameters are shared. Furthermore, the method still inherits the known drawbacks of reverse KL, such as a tendency for mode-collapse, and its overall robustness could be better demonstrated on more complex benchmark tasks.

**Quality:**

3

**Strengths And Weaknesses:**

**Strengths**

- The paper provides a new perspective to understand the connection between reverse KL and log variance.
- The rKL-LD loss provides more stable training behaviour than log-variance and rKL with the reparametrisation trick (rKL-R).

**Weaknesses**

- **Unaddressed Limitations of Reverse KL:** The paper rightly focuses on the benefits of rKL-LD, but it doesn't fully address the known drawbacks. The reverse KL divergence is inherently mode-seeking and prone to mode collapse, a problem the authors acknowledge rKL-LD is still susceptible to. Similarly, the log-derivative trick typically produces higher variance gradients than the reparameterization trick. A deeper analysis of why rKL-LD manages to outperform other rKL methods, despite these issues, would strengthen the paper
- **Apparent Contradiction in Gradient Equivalence:** A central weakness lies in the claimed equivalence between the gradients of rKL-LD and LV loss (Equations 12-13). This equivalence holds only when the forward process parameters are not learned. However, in models like CMCD, the forward and backward processes share the same network parameters. This means the gradient of rKL-LD is *not* equivalent to the gradient of LV in the practical CMCD and DBS setup. The authors should clarify this in the paper to avoid misunderstanding.
- based on the equation 12 and 13, the gradient of rKL-LD is equivalent to LV when \phi = v = empty. However, this is generally not true for CMCD. Note that CMCD parametrise the dirft term of the forward and backward using the same network parameters, which means both q and p depends on the same parameters and the gradiet of rKL-LD is not equivalent to the gradient of LV
- **Lack of Clarity in Final Objectives:** To address the ambiguity mentioned above, the paper would be significantly improved by explicitly stating the final training objective functions for each method tested (e.g., CMCD-rKL-LD, DBS-rKL-LD). Including a clear algorithm or pseudo-code detailing the training procedure would further enhance reproducibility and clarify the exact implementation differences
- **Limited Scope of Evaluation:** The experimental validation could be made more compelling by including more challenging benchmark tasks. Evaluating the proposed method on complex energy landscapes, such as the Lennard-Jones potential, would provide a more robust assessment of its capabilities and limits.

---

> ### Author Rebuttal · Authors · 2025-07-31
>
> We thank the reviewer for this very detailed and thoughtful critique of our manuscript.
>
> **W1 Limitations of rKL:**
>
> First, we would like to stress that the potential mode collapse issue applies analogously to the LV loss which yields - in the absence of shared parameters in the forward and backward process, identical gradients as the rKL-LD loss. We agree that this is a central problem and hence we follow other works and evaluate our method with metrics like the sinkhorn distance, MMD, and EMC which provide insights into mode coverage (Blessing et al., 2024). Whether or not  mode collapse occurs depends on many factors - including the choice of the gradient estimator. As our results in Tab. 2 and 3 show rKL-LD yields overall the best results the aforementioned metrics which probe mode coverage.
> The second critique in this point is that the log-derivative trick can have a higher variance than the reparametrization trick. Frequently, it is assumed that this statement holds true in general but this is not the case. Mohamed et al., 2019, section 5.3.2 discuss that it depends on the Lipschitz constant of the loss function. More generally, even if the variance of the log-derivativative trick gradient estimator is larger it is not obvious whether this additional source of exploration is detrimental or beneficial. For example, low-variance gradients as in large batch SGD are known to yield worse generalization (Keskar et al. 2016), Tab. 2 and 3 suggest that rKL-LD is less prone to mode collapse.
>
>
> **W2: Apparent Contradiction in Gradient Equivalence:**
>
> We would like to clarify that there is no contradiction in our formulation. As explicitly stated in the Abstract (line 4 ff.), Introduction (line 46 ff.), Discussion (line 340 ff.), and the paragraph beginning on line 204 ff., the gradient equivalence between rKL-LD and LV (Equations 12–13) is established under the condition that \phi = v = empty, i.e. there are no learnable forward process parameters and no learnable shared parameters between the forward and reverse process.
> Therefore, we explicitly state in our paper that for diffusion bridge samplers such as CMCD, where the forward and backward processes share the same network parameters, this equivalence does not hold. This observation is precisely what motivates our investigation into the performance of the rKL-LD loss. We hope that the references provided in the submitted version of the paper clarify this point and address the reviewer's concerns.
>
> **W3 Lack of Clarity in Objectives:**
>
> We appreciate the reviewer's feedback and acknowledge that implementing a loss function to achieve the desired gradient can indeed be challenging. To address this, we will include pseudocode for the rKL-R, rKL-LD, and LV loss functions in the final version of the manuscript. Below, we provide the pseudocode for implementing the rKL-LD loss.
>
> **Pseudocode:**
>
> ### **Computation of the rKL-LD Loss**
> ### *Averages are always computed over the batch dimension.*
> ---
>
> **Given:**
>
> A batch of diffusion paths $X_{0:T} \sim q_{\alpha, \nu}(X_{0:T})$
> computed via Euler-Maruyama integration (see Eq. 4).
>
> ---
> 1. **Detach gradients:**
> $$
> X^\prime_{0:T} \gets \mathrm{stop\_grad}(X_{0:T})
> $$
> 2. **Compute log-densities:**
> $$
> \log q_{\alpha, \nu}(X^\prime_{0:T}) \quad \text{(Eq. 5)}
> $$
> $$
> \log p_{\phi, \nu}(X^\prime_{0:T}) \quad \text{(Eq. 6)}
> $$
> 3. **Compute control variate:**
> $$
> b = \mathrm{mean} \left[ \log \frac{q_{\alpha, \nu}(X^\prime_{0:T})}{p_{\phi, \nu}(X^\prime_{0:T})} \right]
> $$
> 4. **Compute advantages (detached):**
> $$
> A^* = \mathrm{stop\_grad} \left[ \log \frac{q_{\alpha, \nu}(X^\prime_{0:T})}{p_{\phi, \nu}(X^\prime_{0:T})} - b \cdot \vec{1} \right]
> $$
> 5. **Compute loss:**
> $$
> \mathcal{L}(\alpha, \phi, \nu) = \mathrm{mean} \left[ A^* \odot \log q_{\alpha, \nu}(X^\prime_{0:T}) \right] - \mathrm{mean}  \left[ \log p_{\phi, \nu}(X^\prime_{0:T}) \right]
> $$
> 6. **Backpropagate and update parameters:**
> $$
> (\alpha, \phi, \nu) \leftarrow \mathrm{Adam} \left( \nabla_{(\alpha, \phi, \nu)} \mathcal{L}(\alpha, \phi, \nu) \right)
> $$
>
> **W4 Limited Scope of Evaluation:**
>
> Training diffusion samplers on this benchmark involves the usage of memory expensive GNN architectures, which is why losses have to be used that allow for subsampling over diffusion times steps to improve memory efficiency. Due to that to the best of our knowledge, diffusion bridges have not yet been successfully scaled to the benchmarks you suggested (see, e.g., Rissanen et al., who report that they were unable to make CMCD work on this benchmark). Nevertheless, we have extended our experimental evaluation to include multimodal benchmarks in higher dimensions such as GMM-100D and MoS-100D. Our results indicate that the rKL-LD loss performs comparably to the LV loss in terms of Sinkhorn distance and Entropic Mode Coverage (EMC), but outperforms the LV loss in terms of Evidence Lower Bound (ELBO), log Z, and Maximum Mean Discrepancy (MMD). On GMM-100D, the LV loss combined with CMCD exhibits divergent behavior in a few seed runs, which significantly degrades its average performance metrics. This problem also occured in MoS-100D for DBS trained with the LV loss. Notably, we again observe that learnable diffusion coefficients improve performance when trained with rKL-LD but worsen performance when trained with LV.
>
> We would like to highlight the comprehensiveness of our experimental evaluation. Specifically, we tested 5 different loss functions across 5 Bayesian learning tasks and 4 multimodal benchmarks, each assessed using 2 distinct diffusion bridge samplers. For each configuration, we performed grid search followed by final run evaluations. In total, this resulted in over 1,350 individual experiments (calculated as 5 × 9 × 2 × 12 + 5 × 9 × 2 × 3). This level of experimental rigor aligns with the standard protocol commonly adopted in published work within this domain (Chen et al.).
>
> We thank you again for your time and effort used to review our work and hope that all of our answers properly address your questions.
>
> **Rissanen, S., OuYang, R., He, J., Chen, W., Heinonen, M., Solin, A., & Hernández-Lobato, J. M. (2025). Progressive Tempering Sampler with Diffusion. arXiv preprint arXiv:2506.05231.**
>
> **Junhua Chen, Lorenz Richter, Julius Berner, Denis Blessing, Gerhard Neumann, and Anima Anandkumar. Sequential controlled langevin diffusions. arXiv preprint arXiv:2412.07081, 2024a.**
>
> **Blessing, Denis, et al. "Beyond ELBOs: A large-scale evaluation of variational methods for sampling." arXiv preprint arXiv:2406.07423 (2024).**
>
> **Mohamed, Shakir, et al. "Monte Carlo gradient estimation in machine learning. arXiv e-prints, page." arXiv preprint arXiv:1906.10652 (2019).**
>
> **Keskar, Nitish Shirish, et al. "On large-batch training for deep learning: Generalization gap and sharp minima." arXiv preprint arXiv:1609.04836 (2016).**
>
> | GMM 100-D | Sinkhorn | ELBO | log Z | EMC | MMD |
> |----------|----------|----------|----------|----------|----------|
> | CMCD: rKL-R $\Large \star$ | $7152.88\pm 177.312$    | $-795.09\pm 6.491$    | $-528.09\pm 2.995$     | $0.995\pm 0.000$     | $0.116\pm 0.001$     |
> | CMCD: LV $\Large \star$   | $22237.03\pm 10627.664$    | $-1957894528.00\pm 1354142383.491$    | $-774.91\pm 466.647$   | $0.866\pm 0.084$    | $0.268\pm 0.101$    |
> | CMCD: LV  | $12420.83\pm 6668.17$   | $-2141430400.00\pm 2031539132.986$    | $-1037.09\pm 931.118$    | $0.951\pm 0.043$    | $0.138\pm 0.015$    |
> | CMCD: rKL-LD $\Large \star$   | $6079.38\pm 205.248$     | $-45.10 \pm 0.091$    | $-18.09\pm 0.341$    |  $\mathbf{0.996 \pm 0.000}$   | $0.084\pm 0.001$    |
> | CMCD: rKL-LD    | $\mathbf{6043.26\pm 166.427}$     | $\mathbf{-45.84\pm 0.08}$    | $\mathbf{-18.88\pm 0.394}$   | $\mathbf{0.996\pm 0.000}$     | $\mathbf{0.085\pm 0.001}$    |
>
> | GMM 100-D | Sinkhorn | ELBO | log Z | EMC | MMD |
> |----------|----------|----------|----------|----------|----------|
> | DBS: rKL-R $\Large \star$    | $5637.36 \pm 155.953$     | $-270.61 \pm 0.275$    | $-138.60\pm 0.85$    | $\mathbf{0.997\pm 0.001}$ | $0.179\pm 0.000$   |
> | DBS: LV $\Large \star$  | $\mathbf{5024.58 \pm 171.079}$     | $-230.83\pm 0.151$    |$-105.53\pm 0.972$    | $\mathbf{0.998\pm 0.001}$    | $0.178\pm 0.000$    |
> | DBS: LV   | $\mathbf{4938.60\pm 103.434}$   | $-252.56\pm 0.299$    | $-116.76\pm 1.061$   | $\mathbf{0.998\pm 0.001}$    |  $0.184\pm 0.001$    |
> | DBS: rKL-LD $\Large \star$   | $\mathbf{5163.31\pm 206.48}$     | $-225.99 \pm 0.132$     | $-101.80\pm 0.953$    | $\mathbf{0.997\pm 0.001}$     | $0.178 \pm 0.000$    |
> | DBS: rKL-LD   | $\mathbf{5132.65\pm 124.068}$   | $\mathbf{-198.53\pm 0.127}$    | $\mathbf{-87.59\pm 0.818}$    | $\mathbf{0.997\pm 0.001}$     | $\mathbf{0.171\pm 0.000}$     |
>
>
> | MoS 100 D | Sinkhorn | ELBO | log Z | EMC | MMD |
> |----------|----------|----------|----------|----------|----------|
> | CMCD: rKL-R $\Large \star$ | $1809.48\pm 298.512$	| $-114.86\pm 0.021$	| $-68.25\pm 0.48$ 	| $0.977\pm 0.004$ 	| $0.265\pm 0.001$ 	|
> | CMCD: LV $\Large \star$	| $2716.16\pm 808.633$	| $-97.44\pm 0.622$	| $-66.37\pm 0.404$   | $\mathbf{0.981\pm 0.005}$	| $0.286\pm 0.001$	|
> | CMCD: rKL-LD $\Large \star$  | $\mathbf{1852.13 \pm 271.333}$ 	| $\mathbf{-68.21\pm 0.322}$	| $\mathbf{-31.71\pm 0.173}$	|  $0.957\pm 0.007$   |$\mathbf{0.260\pm 0.001}$	|
>
> | MoS 100 D | Sinkhorn | ELBO | log Z | EMC | MMD |
> |----------|----------|----------|----------|----------|----------|
> | DBS: rKL-R $\Large \star$ | $9732.62\pm 3761.867$	| $-183.53\pm 0.867$	| $-113.88\pm 1.51$ 	| $0.990\pm 0.003$ 	| $0.298\pm 0.001$ 	|
> | DBS: LV $\Large \star$  | $1964053495808.00 \pm 783156518610.415$ 	| $-168435.12 \pm 66865.384 $	|$-48409.64 \pm 18962.519 $	| $0.393\pm 0.201$	| $0.298\pm 0.001$	|
> | DBS: rKL-LD $\Large \star$   | $\mathbf{3626.16\pm 1557.097}$ 	| $\mathbf{-121.77 \pm 0.926}$ 	| $\mathbf{-69.53 \pm 0.347}$	| $\mathbf{0.996 \pm 0.001}$ 	| $\mathbf{0.295 \pm 0.001}$	|

---

> > ### Comment · Reviewer_KBTa · 2025-08-04
> >
> > Thanks for the detailed response. I will increase my score to 4 since most of my concerns have been addressed. It would be great to see the authors include the pseudocode in the paper

---

### Official Review · Reviewer_ET35 · 2025-06-30

**Clarity:** 3
**Significance:** 3
**Originality:** 2
**Rating:** 5
**Confidence:** 3

**Summary:**

- Examines the losses used to train diffusion bridges and controlled monte carlo diffusion where both the forwards and reverse process are parameterized.
- Examine potential issues with the log variance (LV) loss typically used, focusing on stability and adherence to the data processing inequality
- Argue for the application of the rKL loss with the log-derivative trick
- Demonstrates empirical gains with their proposed rKL-LD loss when applied to both CMCD and DBS training when compared to LV and rKL-R loss
- Additionally, show that rKL-LD allows for further improvements when combined with learnable diffusion parameters, which tend to destabilize training with LV loss

**Questions:**

In Table 2, it seems that rKL-LD provides significant benefits over LV when applied to CMCD for the GMM-40 task. Furthermore, the ability to learn the diffusion parameters also provides additional gains. However, when applied to DBS, it seems that the opposite holds: rKL-LD decreases performance relative to the LV loss, and LV seems to further improve performance with learnable diffusion parameters whereas rKL-LD performance deteriorates. Why does this occur?

What are the practical effects of the LV loss not satisfying the data processing inequality? While I understand this as a theoretical issue, I am not sure if this results in tangible effects. It would be nice if the authors could connect this back to observable phenomena that occur during training.

Overall, I am leaning weakly towards acceptance. I think if the above questions are answered, I will be happy to increase my score.

**Ethical Concerns:**

["NO or VERY MINOR ethics concerns only"]

**Final Justification:**

I initially had concerns regarding the novelty, and the importance of the DPI violation. The authors have thoroughly addressed each part, and thus I feel that there are no other issues with this paper as far as I can see. I think this would be fairly impactful paper for the field of diffusion bridges / diffusion samplers, so I think a 5 is appropriate.

**Limitations:**

I think that it would be nice if there were empirical results on more difficult tasks, such as NICE [1], or alanine dipeptide [2], but the results included are sufficient already.

[1] Score-Based Diffusion meets Annealed Importance Sampling. Doucet et al. NeurIPS 2022
[2] Flow Annealed Importance Sampling Bootstrap. Midgley et al. ICLR 2023.

**Paper Formatting Concerns:**

I did not observe any major formatting issues.

**Quality:**

3

**Strengths And Weaknesses:**

*Quality*

Overall, this is a good quality paper. While the experiments are not very high dimensional, this is typical for the field of diffusion sampling (from what I know), and the authors do include results on LGCP, which is 1600 dimensional. The main point — that LV loss is not suitable for diffusion sampling when both the forward and reverse process is learned — seems fairly clear. The analytical comparison of the gradients for the different losses seems accurate.

*Clarity*

One significant strength of this paper is its coverage of the problem background and prior work. I appreciate the thoroughness of this section.
Overall, the paper is well organized and the writing style is very easy to follow. I do think that the writing style could be improved for the experiments section. In particular, the paragraphs were very long and difficult to follow — it would be better to break them up into smaller paragraphs that focus on specific findings. But I do not find this to be that significant of an issue

*Significance*

The findings of this paper are fairly significant. I agree with the discussion in section 3.3, which discusses the importance of learning diffusion parameters, as different target distributions may require different diffusions to adequately capture the entropy of the target distribution while balancing the signal to noise ratio to ensure stable training.

*Originality*

The biggest weakness of this paper is the originality: as noted by the authors in the related works, the application of the log-derivative trick to the rKL loss has been explored before in the context of discrete diffusion samplers.

While the proposed loss is not new, I think the analysis and the application towards CMCD / DBS with learnable diffusion parameters is novel. Thus I find the overall originality to be fair.

---

> ### Author Rebuttal · Authors · 2025-07-31
>
> We sincerely appreciate your positive evaluation. It’s encouraging to know that our work is perceived as a good quality paper with significant findings and good coverage of the problem background. Please find our detailed responses below.
>
> **Weaknesses:**
>
> **On Clarity:**
>
> Thank you for your suggestion, we improved the writing style in our experiment section as you suggested, i.e. shorter paragraphs that are dedicated to specific findings.
>
> **On Originality:**
>
> The fact the log-derivative trick was employed in prior work for discrete diffusion samplers is regarded as the main weakness of our work. We challenge the associated argumentation by the reviewer since in the mentioned works on discrete diffusions this method is applied to diffusion methods that do not require any backpropagation through the forward diffusion process, i.e. these are not diffusion bridge samplers. For diffusion bridge samplers the application of the log-derivative is thus novel, regardless of whether the considered domain is continuous or discrete.
> Apart from this conceptual difference these discrete samplers are studied on entirely different benchmarks. Importantly, within the significantly larger field of continuous domain samplers the log-derivative is of course well-known but as pointed out in our manuscript it was dismissed for questionable reasons based on the variance of the gradient estimators that do ultimately not have any predictive power over its actual performance compared to the reparametrization trick. One of our novel insights is that the log-derivative trick is indeed very competitive and often even superior, refuting the argumentation in prior works against it. While this is just one of our contributions, we still think that it is a genuinely new and highly important insight that challenges the predominant choice of gradient estimators and losses in this field.
>
> **Questions:**
>
> **On Results in Table 2:**
>
> We would like to note that these performance differences w.r.t. Sinkhorn are not statistically significant (hence multiple results are bold) whereas significant performance improvements can be achieved by rKL-LD with learned diffusion parameters w.r.t. ELBO and MMD. Hence, in the metrics that yields a single best method on GMM40 with statistical significance, we always find that rKL-LD performs best.
>
> **Impact of DPI violation**
>
> It is beyond the scope of this paper to empirically investigate practical drawbacks of the LV loss that stem from its DPI violation. However, in our response to Weakness 1 of reviewer iCJN we provide a more compelling counterexample that shows that the LV loss violates the DPI. What this new example illustrates is that unless the learned distribution is absolutely continuous with respect to the target distribution, it is possible that the LV loss on the joints reaches its global minimum, i.e. it vanishes, while the corresponding marginal distributions are not equal. In this situation the LV loss on the joint distribution cannot provide any insights into the divergence between the marginals which are the ultimate target of these samplers.  In practice, this situation cannot be excluded due to finite numerical precision. While this is still a theoretical argument, we hope that it supports our scepticism that the LV loss is a reliable loss for diffusion bridge samplers.
>
> **Limitations**
>
> **On More Difficult Tasks:**
>
> Due to the limited time of the rebuttal period we had to make a compromise on which additional benchmarks to address to satisfy all the reviewers' requests (see reviewer iCJN's request). We chose to expand our experimental evaluation with GMM 100-D and MoS 100-D below. The results support once more the strength of our rKL-LD-based methods.
>
> | GMM 100-D | Sinkhorn | ELBO | log Z | EMC | MMD |
> |----------|----------|----------|----------|----------|----------|
> | CMCD: rKL-R $\Large \star$ | $7152.88\pm 177.312$    | $-795.09\pm 6.491$    | $-528.09\pm 2.995$     | $0.995\pm 0.000$     | $0.116\pm 0.001$     |
> | CMCD: LV $\Large \star$   | $22237.03\pm 10627.664$    | $-1957894528.00\pm 1354142383.491$    | $-774.91\pm 466.647$   | $0.866\pm 0.084$    | $0.268\pm 0.101$    |
> | CMCD: LV  | $12420.83\pm 6668.17$   | $-2141430400.00\pm 2031539132.986$    | $-1037.09\pm 931.118$    | $0.951\pm 0.043$    | $0.138\pm 0.015$    |
> | CMCD: rKL-LD $\Large \star$   | $6079.38\pm 205.248$     | $-45.10 \pm 0.091$    | $-18.09\pm 0.341$    |  $\mathbf{0.996 \pm 0.000}$   | $0.084\pm 0.001$    |
> | CMCD: rKL-LD    | $\mathbf{6043.26\pm 166.427}$     | $\mathbf{-45.84\pm 0.08}$    | $\mathbf{-18.88\pm 0.394}$   | $\mathbf{0.996\pm 0.000}$     | $\mathbf{0.085\pm 0.001}$    |
>
> | GMM 100-D | Sinkhorn | ELBO | log Z | EMC | MMD |
> |----------|----------|----------|----------|----------|----------|
> | DBS: rKL-R $\Large \star$    | $5637.36 \pm 155.953$     | $-270.61 \pm 0.275$    | $-138.60\pm 0.85$    | $\mathbf{0.997\pm 0.001}$ | $0.179\pm 0.000$   |
> | DBS: LV $\Large \star$  | $\mathbf{5024.58 \pm 171.079}$     | $-230.83\pm 0.151$    |$-105.53\pm 0.972$    | $\mathbf{0.998\pm 0.001}$    | $0.178\pm 0.000$    |
> | DBS: LV   | $\mathbf{4938.60\pm 103.434}$   | $-252.56\pm 0.299$    | $-116.76\pm 1.061$   | $\mathbf{0.998\pm 0.001}$    |  $0.184\pm 0.001$    |
> | DBS: rKL-LD $\Large \star$   | $\mathbf{5163.31\pm 206.48}$     | $-225.99 \pm 0.132$     | $-101.80\pm 0.953$    | $\mathbf{0.997\pm 0.001}$     | $0.178 \pm 0.000$    |
> | DBS: rKL-LD   | $\mathbf{5132.65\pm 124.068}$   | $\mathbf{-198.53\pm 0.127}$    | $\mathbf{-87.59\pm 0.818}$    | $\mathbf{0.997\pm 0.001}$     | $\mathbf{0.171\pm 0.000}$     |
>
>
> | MoS 100 D | Sinkhorn | ELBO | log Z | EMC | MMD |
> |----------|----------|----------|----------|----------|----------|
> | CMCD: rKL-R $\Large \star$ | $1809.48\pm 298.512$	| $-114.86\pm 0.021$	| $-68.25\pm 0.48$ 	| $0.977\pm 0.004$ 	| $0.265\pm 0.001$ 	|
> | CMCD: LV $\Large \star$	| $2716.16\pm 808.633$	| $-97.44\pm 0.622$	| $-66.37\pm 0.404$   | $\mathbf{0.981\pm 0.005}$	| $0.286\pm 0.001$	|
> | CMCD: rKL-LD $\Large \star$  | $\mathbf{1852.13 \pm 271.333}$ 	| $\mathbf{-68.21\pm 0.322}$	| $\mathbf{-31.71\pm 0.173}$	|  $0.957\pm 0.007$   |$\mathbf{0.260\pm 0.001}$	|
>
> | MoS 100 D | Sinkhorn | ELBO | log Z | EMC | MMD |
> |----------|----------|----------|----------|----------|----------|
> | DBS: rKL-R $\Large \star$ | $9732.62\pm 3761.867$	| $-183.53\pm 0.867$	| $-113.88\pm 1.51$ 	| $0.990\pm 0.003$ 	| $0.298\pm 0.001$ 	|
> | DBS: LV $\Large \star$  | $1964053495808.00 \pm 783156518610.415$ 	| $-168435.12 \pm 66865.384 $	|$-48409.64 \pm 18962.519 $	| $0.393\pm 0.201$	| $0.298\pm 0.001$	|
> | DBS: rKL-LD $\Large \star$   | $\mathbf{3626.16\pm 1557.097}$ 	| $\mathbf{-121.77 \pm 0.926}$ 	| $\mathbf{-69.53 \pm 0.347}$	| $\mathbf{0.996 \pm 0.001}$ 	| $\mathbf{0.295 \pm 0.001}$	|
>
> We thank you again for your time and effort used to review our work. If you are satisfied with our answers and extensions, we kindly ask you to take this into account in your final decision.

---

> > ### Comment · Reviewer_ET35 · 2025-08-02
> > **Response to Rebuttal**
> >
> > Thank you for your rebuttal. I will respond to your points below.
> >
> > * **On Originality**
> >
> > I think your argument for this point makes sense -- while previous works have investigated the same loss, this is the first work to study it in the context of diffusion bridges. Thus I do not find this to be a weakness with this submission anymore.
> >
> > * **Results Table 2**
> >
> > This clarification makes the results much more understandable and aligned with the overall narrative.
> >
> > * **Impact of DPI Violation**
> >
> > I am not sure why the practical impacts of the DPI violation for LV loss is beyond the scope of this paper. In the rebuttal, the author's state:
> >
> > > It is beyond the scope of this paper to empirically investigate practical drawbacks of the LV loss that stem from its DPI violation
> >
> > In lines 147-149, the paper states:
> >
> > > We therefore argue that the data processing inequality is an essential property for losses
> > > in sampling setting and that losses that do not provide such a conceptual footing might be practically
> > > problematic.
> >
> > Given that the lack of a conceptual footing is explicitly argued to be a source of practical issues, I am simply asking what those issues are. This seems like a natural question in response to the authors’ own claim.
> >
> > Furthermore, I do not find the provided counter-example compelling: this seems to be an issue with absolute continuity, not violation of DPI. The counterexample defines a $q(x, y)$ that is not absolutely continuous w.r.t $p(x, y)$. Even though such scenarios may arise in practice, absolute continuity is foundational for the theory of diffusion modeling.
> >
> > From what I understand, the assumption of absolute continuity is required for the existence of the reverse diffusion process -- the reverse diffusion process is an application of Girsanov's theorem [1], which requires the path measure of the learned process to be absolutely continuous w.r.t to the target SDE. If you look at [2], the theorem requires satisfaction of Novikov's condition, which is a sufficient condition for absolute continuity [3].
> >
> > While the provided counter-example demonstrates an issue with LV losses, it does so in a setting where most diffusion-based frameworks would become ill-posed, especially any that require the existence of a reverse process. Thus this counter-example falls outside the regime of assumptions typical for diffusion models.
> >
> >
> > [1]. Maximum Likelihood Training of Score-Based Diffusion Models. Song et al. NeurIPS 2021.
> >
> > [2]. Stochastic differential equations, Chapter 8.6. Øksendal B. 2003.
> >
> > [3]. On conditions for absolute continuity of probability measures. Novikov 1978.
> >
> >
> > * **Experimental Results**
> >
> > Thank you for extending the experiments to larger scale MoG and MoS. From what I understand, these seem to be easier distributions than the LGCP experiment already included, which is 1600d, in comparison to the new 100d experiments. Furthermore, I do not think that these are at the level of difficulty as NICE or Alanine Dipeptide, which are the experiments I suggested.
> >
> > However, I do not count this as a negative -- I completely understand and empathize with the very quick turnaround for the rebuttal period. I mentioned these experiments as things that may be considered for future work or extensions. I found the experiments in general to be sufficient to begin with.
> >
> > **Overall**
> > My only remaining point of concern is the DPI violation -- if the authors can demonstrate that this results in practical consequences, or provide a counter-example that does not break absolute continuity but demonstrates the necessity of DPI, then I will raise my score.

---

> > > ### Author Response · Authors · 2025-08-03
> > >
> > > We sincerely thank the reviewer for the thoughtful and detailed feedback, and we greatly appreciate the acknowledgement of the paper’s originality and the clarity improvements to Table 2 and experimental results.
> > >
> > > **Absolute Continuity in the Triangle-Counterexample**
> > >
> > > The reviewer states: "The counterexample defines a $q(x,y)$ that is not absolutely continuous w.r.t $p(x,y)$.". We disagree with the reviewer here and we state again our counterexample:
> > >
> > > *   A reference distribution $p(x,y)=1$ on the full unit square $\Omega =\[0,1\]^2$, and
> > >
> > > *   A learned distribution $q(x,y)=2$ on the triangle $\{(x,y)∈\[0,1\]^2∣y≤1−x}$, and $0$ elsewhere.
> > >
> > > By definition, q≪p if every measurable set $A⊆\[0,1\]^2$ for which $p(A)\=0$ also satisfies $q(A)\=0$.
> > >
> > > This is satisfied in our case because:
> > >
> > > *   The support of $q(x,y)$ (the triangle) lies entirely within the support of $p(x,y)$ (the full square).
> > >
> > > *   The only sets with zero measure under $p(x,y)$ are outside of $\Omega$, and $q(x,y)$ also assigns zero mass to those.
> > >
> > > **Continuous Counterexample with Identical Supports**
> > >
> > > To demonstrate indisputably that the violation of the DPI with the LV-loss does not depend on absolute continuities we design a modified counterexample in which the distributions have identical supports. Consider:
> > >
> > > *   A reference distribution $p(x,y)\=1$ on the full unit square $\Omega=\[0,1\]^2$, and
> > >
> > > *   A learned distribution $ q(x, y)$ on the unit square $\[0,1]^2$ as:
> > >
> > > \\[
> > > q(x, y) =
> > > \\begin{cases}
> > > \\frac{2n}{n + 1} & \\text{if } x + y < 1 \\\\
> > > \\frac{2}{n + 1} & \\text{if } x + y \\ge 1
> > > \\end{cases}
> > > \\]
> > >
> > > Thus, the density of $ q(x, y)$ in the lower-left half of the unit square is $n$ times the density in the upper-right half.
> > >
> > > For $n=10^{-3}$ this example yields via an analytic calculation the following violation of the DPI:
> > >
> > > *  For the joints: $var_{q(x,y)} \log \frac{q(x,y)}{p(x,y)} \approx 0.0476218$
> > >
> > > *  For the marginals: $var_{q(x)} \log \frac{q(x)}{p(x)} \approx 0.25$
> > >
> > > Note that for $n=0$ this yields an example that is essentially equivalent to the original continuous conterexample.
> > >
> > > We hope that this new counterexample with $n=10^{-3}$ convinces the author that the violation of the DPI is relevant in the context of diffusion samplers.

---

> > > > ### Comment · Reviewer_ET35 · 2025-08-04
> > > >
> > > > Thank you for your response. This satisfies my concerns regarding the violation of DPI. I also think this partially resolves my concerns regarding the practical consequences of DPI violation. I will increase my score to reflect this.

---

### Official Review · Reviewer_Ce3h · 2025-07-02

**Clarity:** 4
**Significance:** 3
**Originality:** 3
**Rating:** 5
**Confidence:** 3

**Summary:**

In this paper the problem setup considered is learning to sample from a data distribution $\pi_0$ with no available samples/observations from $\pi_0$. For this problem setup, different loss functions are contrasted and compared: namely, the reverse Kullback-Leibler (rKL) loss or the log variance (LV) loss.  In prior works, the LV loss has performed better than the rKL loss but the paper argues this is due to the application of the reparametrisation trick when computing the rKL loss and instead motivates using the rKL loss with the log derivative trick.

**Questions:**

1. See Weakness 2. above.
2. Can you explain what they mean by “frequent iteration of the parametrization trick” in line 169? Do you mean due to the successive application of Euler-Maruyama to get the values $X_t$ of one trajectory for different values of $t$?
3. Related to Weakness 3 above, can you say anything about the gradients obtained via the two different tricks and why the log derivative trick is expected to outperform the reparametrisation. Or for example, can you quantify the error induced via the reparametrisation trick?
4. Currently, I’m not completely sure what I should take away from the gradients of the rKL loss with the log derivative, other than in specific circumstances (as stated in section 3.2) it coincides with the LV loss. Are there any more conclusions to be drawn for example in relation to the performance of the loss?

**Ethical Concerns:**

["NO or VERY MINOR ethics concerns only"]

**Final Justification:**

My main weakness was comparing the log derivative and reparametrisation trick which has been addressed in the rebuttals.

**Limitations:**

Yes.

**Paper Formatting Concerns:**

None.

**Quality:**

3

**Strengths And Weaknesses:**

Strengths:

1. The paper is well-written and structured and I found it easy to follow with illuminating descriptions of losses through the lens of $f$-divergences. The paper presents concise comparisons between these commonly used losses as well as the log variance loss(es) which are not $f$-divergences.

2. The submission shows that the log variance loss does not satisfy the data processing inequality. I find this to be an important observation when choosing such a loss for diffusion models.

3. The submission shows through experiments that the rKL loss with the log derivative trick performs well in comparison to both the rKL loss with the reparametrisation trick as well as the LV loss. The experiments conducted are the same as those in Chen et al. (2024a).


Weaknesses:

1. One of the main points made in the paper is to advocate for the use of the rKL loss with the log derivative trick. However, the definition of the log derivative trick is relegated to the appendix, alongside the reparametrisation trick. I would find it beneficial to have this in the main paper. Moreover, I would find it beneficial to have more of an explanation of the log derivative trick alongside the definition.

2. Related to this, in A.2 is there a reason for switching to the notation $\langle O(X) \rangle_{X \sim p}$ as opposed to $\mathbb{E}_{X\sim p} [O(X)]$ for the expectation as used in the rest of the paper?

3. The paper motivates using the rKL objective with the log derivative trick instead of the LV loss well. However, I do not understand why it should be expected that the rKL objective with the log derivative trick is expected to do better than the rKL loss with the reparametrisation trick, or how these methods relate to each other. A more concrete comparison would improve the clarity of this work.


To conclude, I think this is a well-written paper and I find the contributions of comparing the loss functions compelling. However, I think there are some weaknesses to be addressed, mostly in terms of a comparison between the reparametrisation trick and the log derivative trick for the gradients. For this reason I currently lean towards a weak accept, but I’m open to revising my score should the weaknesses be easy to resolve.

---

> ### Author Rebuttal · Authors · 2025-07-31
>
> We appreciate the reviewer's thoughtful and constructive feedback. We are also grateful for the acknowledgment of our paper's clarity, the significance of our finding that the LV loss violates the Data Processing Inequality, and the recognition of our experimental results.
>
> **Weaknesses:**
>
> **W1:**
> We agree that both the log-derivative trick and the reparametrization trick play such a central role in this paper that they deserve a self-contained discussion. However, due to the space limitation of 9 pages, we cannot add any further material to the main paper. Hence, we decided to add clear references to sections in the appendix that discuss both methods and provide intuitive explanations of the corresponding computational graphs in the context of the vanishing/exploding gradient problems.
>
> **W2:**
> Thank you for your comment, we changed this in the updated version and stick to notation style $\mathbb{E}_{X \sim p} [O(X)]$.
>
> **W3:**
> Regarding the problems of the rKL-R we mention three times in the paper the vanishing / exploding gradient problem. However, the reviewers statement clarified to us that we did not explicitly point out that this problem does not arise to the same extent with the log-derivative trick.
> We will explain the underlying reason for this in detail in the newly added appendix section that we mentioned in our response to W1. Here we will try to give a brief outline of this reason.
> For the reparametrization trick, backpropagation propagates through the consecutively generated samples $X_t$ along the diffusion path from the final sample ($X_0$) all the way back to the initial sample from the prior $X_T$. The corresponding paths taken by the backpropagation pass through the computational graph of the entire diffusion process with all of its steps, e.g. T = 128 in our case. These long paths can result in the vanishing / exploding gradients problem. Indeed, as mentioned in our manuscript, prior work points out this issue specifically in the context of diffusion samplers. See in particular the discussion in Vargas, et al., 2023 appendix C. 8.1 where it is shown that adding additional stop_gradient operators helps in alleviating this problem, at the cost of introducing biased gradient estimates. This problem does not arise for the log-derivative trick since it does not involve gradient flow through the consecutive sampling processes in diffusion samplers and is thus not subject to this problem. Details on the involved computational graphs will be provided in the updated appendix.
> We would like to stress, however, that this difference between the reparametrization trick and log-derivative trick or considerations of the variance of these gradient estimators (see first weakness of reviewer KBTa) are not sufficient to argue that one is better than the other. In general, the success of non-convex optimization depends on trade-offs in terms of exploration and exploitation. Evidently it is hard to argue theoretically whether methods that boost exploration or exploitation are generally preferable. Consequently, empirical investigations play a central role in addressing these questions. One of our contributions is that we are the first to empirically investigate the log-derivative trick for diffusion bridge samplers in continuous domains and we find that it does outperform popular methods.
>
> **Questions:**
>
> **Q1:**
>  See our response to W2 above.
>
> **Q2:**
> Thanks for pointing this out; it is a typo. It will be corrected to refer to the “reparametrization trick”. Here we refer to the issue addressed in W3 above, which is associated with long paths of the backpropagation algorithm through the computational graph that is generated by repeated Euler-Maruyama steps, i.e., by the consecutive diffusion steps. Also, here we will point the reader to the aforementioned new appendix on the issue with the reparametrization trick.
>
> **Q3:**
> Frequently, the reparametrization trick is argued in the literature to have a lower variance in the gradient estimation than the log-derivative trick. However, this is only true if the Lipschitz constant of the loss is sufficiently small (Mohamed et al, 2019 section 5.3.2). Arguably, it is, however, not necessarily the case that a low-variance gradient estimator outperforms a high-variance estimator. The reason is that in non-convex optimization the trade-off between exploration and exploitation is key. Hence, higher-variance gradient estimators could contribute to more exploration and hence be beneficial. Thus investigating the gradient variances cannot provide definitive insights into why one gradient estimator is better than another. For an illustration of the potentially detrimental effects of lower variance gradient estimators, see the influential work by Keskar et al., 2016.
>
> We appreciate your time and effort in reviewing our work. If our responses and revisions meet your expectations, we kindly ask that you consider them in your final decision.
>
> **Vargas, Francisco, Will Grathwohl, and Arnaud Doucet. "Denoising diffusion samplers." arXiv preprint arXiv:2302.13834 (2023).**
>
> **Mohamed, Shakir, et al. "Monte Carlo gradient estimation in machine learning. arXiv e-prints, page." arXiv preprint arXiv:1906.10652 (2019).**
>
> **Keskar, Nitish Shirish, et al. "On large-batch training for deep learning: Generalization gap and sharp minima." arXiv preprint arXiv:1609.04836 (2016).**

---

> > ### Comment · Reviewer_Ce3h · 2025-08-04
> >
> > Thank you for your detailed response and discussion. Including this extra discussion (W3) in the paper addresses my concerns and I recommend for the paper to be accepted.

---

### Decision · Program_Chairs · 2025-09-17

**Decision:**

Accept (poster)

**Comment:**

This paper studied two loss functions used in literature for training diffusion samplers: log variance and reverse KL losses. It advocates for using the reverse KL loss with the log-derivative trick (rKL-LD), particularly for diffusion bridges where both forward and reverse processes are learned. The conclusion goes against an earlier work that shows log variance loss outperforms rKL loss - which the author pointed out to be an artifact of using the reparametrization trick for optimization.

All reviewers vote for acceptance, highlighting the significance of the finding, the paper's excellent clarity, and the empirical gains. Several concerns regarding equivalence results and counter examples was sufficiently addressed during rebuttal. Therefore, I recommend acceptance.